



# Hotspots and drivers of compound marine heatwave and low net primary production extremes

Natacha Le Grix[1,2], Jakob Zscheischler[1,2,3], Keith B. Rodgers[4,5], Ryohei Yamaguchi[4,5], and Thomas L. Frölicher[1,2]

[1]Climate and Environmental Physics, Physics Institute, University of Bern, Bern, Switzerland
[2]Oeschger Centre for Climate Change Research, University of Bern, Bern, Switzerland
[3]Department of Computational Hydrosystems, Helmholtz Centre for Environmental Research – UFZ, Leipzig, Germany
[4]Center for Climate Physics, Institute for Basic Science, Busan, South Korea
[5]Pusan National University, Busan, South Korea

**Correspondence:** Natacha Le Grix (natacha.legrix@unibe.ch)

**Abstract.** Extreme events can severely impact marine organisms and ecosystems. Of particular concern are multivariate compound events, namely when conditions are simultaneously extreme for multiple ocean ecosystem stressors. In 2013-2015 for example, an extensive marine heatwave (MHW), known as the Blob, co-occurred locally with extremely low net primary productivity (NPPX) and negatively impacted marine life in the northeast Pacific. Yet, little is known about the characteristics and

drivers of such multivariate compound MHW-NPPX events. Using five different satellite-derived NPP estimates and large ensemble simulation output of two widely-used and comprehensive Earth system models, GFDL-ESM2M-LE and CESM2-LE, we assess the present-day distribution of compound MHW-NPPX events and investigate their potential drivers on the global scale. The satellite-based estimates and both models reveal hotspots of frequent compound events in the center of the equatorial Pacific and in the subtropical Indian Ocean, where their occurrence is at least three times higher (more than 10 days per year)

than if MHWs (temperature above the 90th percentile threshold) and NPPX events (NPP below the 10th percentile threshold) were to occur independently. However, the models show disparities in the northern high latitudes, where compound events are rare in the satellite-based estimates and GFDL-ESM2M-LE (less than 3 days per year), but relatively frequent in CESM2-LE. In the Southern Ocean south of 60°S, low agreement between the observation-based estimates makes it difficult to determine which of the two models better simulates MHW-NPPX events. The frequency patterns can be explained by the drivers of com-

pound events, which vary among the two models and phytoplankton types. In the low latitudes, MHWs are associated with enhanced nutrient limitation on phytoplankton growth, which results in frequent compound MHW-NPPX events in both models. In the high latitudes, NPPX events in GFDL-ESM2M-LE are driven by enhanced light limitation, which rarely co-occurs with MHWs, resulting in rare compound events. In contrast, in CESM2-LE, NPPX events in the high latitudes are driven by reduced nutrient supply that often co-occurs with MHWs, moderates phytoplankton growth and causes biomass to decrease.

Compound MHW-NPPX events are associated with a relative shift towards larger phytoplankton in most regions, except in the eastern equatorial Pacific in both models, as well as in the northern high latitudes and between 35°S and 50°S in CESM2-LE, where the models suggest a shift towards smaller phytoplankton, with potential repercussions on marine ecosystems. Overall, our analysis reveals that the likelihood of compound MHW-NPPX events is contingent on model representation of the fac-





tors limiting phytoplankton production. This identifies an important need for improved process understanding in Earth system
models used for predicting and projecting compound MHW-NPPX events and their impacts.

## 1   Introduction

Warming and reduced primary productivity of organic matter by marine phytoplankton are considered to be two of the major
potential stressors of open ocean ecosystems, along with acidification and deoxygenation (Gruber, 2011; Bopp et al., 2013;
Bindoff et al., 2019). Marine ecosystems are not only threatened by long-term decadal-scale changes in sea surface temper-
ature (SST) (Cheng et al., 2017) and net primary productivity (NPP) (Boyce et al., 2010; Doney et al., 2012), they are also
increasingly impacted by short-term extreme events, such as marine heatwaves (MHWs) (Wernberg et al., 2013; Frölicher and
Laufkötter, 2018; Oliver et al., 2018) and extremely low NPP events (hereafter called 'NPPX' events; Whitney (2015); Cav-
ole et al. (2016)). An emerging concern is the occurrence of multivariate compound events, namely situations when multiple
ecosystem stressors deviate from normal conditions simultaneously, in close spatial proximity or temporal succession (Leonard
et al., 2014; Zscheischler et al., 2018, 2020). Together they may severely impact marine ecosystems (Boyd and Brown, 2015;
Gruber et al., 2021). To date, the majority of studies have focused on compound events over land (e.g. Ridder et al. (2020);
Zscheischler et al. (2020)), with only a relatively small number of studies having addressed compound events in the ocean
(Gruber et al., 2021; Shi et al., 2021; Le Grix et al., 2021; Mogen et al., 2022; Burger et al., 2021).

The combination of MHW and NPPX may cause severe impacts on marine organisms and ecosystems (Boyd and Brown,
2015; Le Grix et al., 2021). 'The Blob' in the Northeast Pacific stands as an example of such an impactful compound event.
Between 2013 and 2015, the Northeast Pacific experienced the most intense and longest-lasting MHW ever recorded, with
maximum SST anomalies of more than 5°C lasting for more than 350 days (Di Lorenzo and Mantua, 2016; Laufkötter et al.,
2020). It coincided with large negative anomalies in phytoplankton NPP (Whitney, 2015) associated with reduced coastal
upwelling, and it had severe impacts for marine life (Cavole et al., 2016), including extreme mortality and reproductive failure
of sea birds (Jones et al., 2018; Piatt et al., 2020), mass strandings of whales in the western Golf of Alaska and of sea lions in
California, not to mention shifts in species distribution towards warm-water species (Cavole et al., 2016; Cheung and Frölicher,
2020). Although not all compound MHW and NPPX events may lead to extreme consequences for marine organisms and
ecosystems, they should at the very least be considered as compound hazards (Ridder et al., 2022), and as such, pose a threat
that warrants further investigation.

In a previous study, Le Grix et al. (2021) characterized compound high SST and low chlorophyll events, with low chlorophyll
assumed as a proxy for low phytoplankton biomass. Using satellite-derived chlorophyll and SST observations, they found
hotspots of frequent compound events in the equatorial Pacific, in the Indian Ocean and along the borders of the subtropical
gyres. In these regions, more than 10 compound event days occur per year. This is 3 to 7 times more often than expected under
the assumption of independence between high SST and low chlorophyll events. The authors also showed that compound event
occurrence is strongly modulated over interannual time-scales by large-scale modes of climate variability. An example is the
El Niño-Southern Oscillation, whose positive phase is associated with increased occurrence of compound events in the eastern





equatorial Pacific. Although the state of climate modes provides valuable information regarding the likelihood of compound events to occur, much remains to be learned regarding local physical and biological drivers of such compound events. Enhanced mechanistic understanding of these potentially harmful events in the ocean is crucial for building and improving the tools for

their prediction and ultimately for adaptation and ecosystem management (Gruber et al., 2021).

Previous studies have investigated the drivers of MHWs, which can act on various spatial and temporal scales (e.g., Holbrook et al. (2019); Gupta et al. (2020); Oliver et al. (2021); Vogt et al. (2022)). MHWs can be triggered through local processes affecting the temperature budget of the mixed layer such as air-sea heat fluxes, local vertical mixing or advection (Gupta et al., 2020; Vogt et al., 2022), while MHWs can also be caused remotely through atmospheric or oceanic teleconnection processes

(Bond et al., 2015; Holbrook et al., 2019). A number of studies have investigated phytoplankton variability using data derived from satellite ocean color (Boyce et al., 2010; Whitney, 2015; Gittings et al., 2018; Long et al., 2021a). However, only a few studies have explored the drivers of NPPX events during MHWs. For example, Whitney (2015) show that in winter 2013/2014 during the 'The Blob' anomalous winds weakened nutrient transport to the northeastern Pacific transition zone and decreased phytoplankton NPP, resulting in the lowest chlorophyll concentrations ever measured. Wyatt et al. (2022) suggest that nutrient

limitation during MHWs generally reduces the biomass of small and large phytoplankton in the northeast Pacific transition zone. However, not all warming events are accompanied by NPPX events. For instance, Long et al. (2021a) noted an increase in NPP during two recent MHWs in the Northeast Pacific. Even though high SST may be associated with nutrient limitation on phytoplankton growth and with enhanced phytoplankton grazing, it also directly enhances phytoplankton growth (Laufkötter et al., 2015). Phytoplankton is indeed modulated by multiple interacting processes in the ocean, rendering it a complex task to

identify drivers of any extreme change in NPP. As data derived from satellite observations can be sparse, biased or uncertain (Behrenfeld et al., 2005; Long et al., 2021a) and limited to recent decades, multiple simulations from Earth system models that include a biological component in the ocean appear as a useful tool to improve our lack of understanding of NPP variability and extremes.

Extreme events are rare by definition and compound extreme events occur even less frequently. Understanding compound

MHW-NPPX events from a statistical point of view requires therefore large datasets from which to sample numerous combinations of extremely high SST and extremely low NPP. Over our period of interest (e.g., satellite period 1998-2018) both extremes rarely co-occur together. In this context, large ensemble simulations (LES) with climate models (Frölicher et al., 2009; Deser et al., 2020) provide an invaluable tool for advancing our understanding of compound events. LES are created with a single climate model under a particular historical or future radiative forcing scenario, by applying perturbations to the

initial conditions of each member in order to create diverging climate trajectories. LES provide the necessary large datasets from which to infer the uncertainty in the likelihood of compound events. Uncertainty from internal variability can be inferred from one ensemble's members spread, whereas model differences are assessed by comparing LES forced by different models. Here, we use LES from two global coupled climate Earth System Models, GFDL's ESM2M and CESM2, to investigate compound MHW-NPPX events.

The principal objectives of our study are to identify hotspots of compound MHW-NPPX events, to assess the fidelity of both Earth system models in simulating MHW-NPPX events, and to gain mechanistic insights into processes driving these





compound events, to thereby enhance our capacity to better project the occurrence of such events into the future. We focus on the satellite period (1998-2018) over which we have satellite-based data of NPP.

## 2 Methods

### 2.1 Observation-based data

We use SST data from NOAA's daily high-resolution Optimum Interpolation SST (OISST) analysis product with a horizontal resolution of 0.25° latitude x 0.25° longitude (Reynolds et al., 2007; Banzon et al., 2016). This observation-based data product provides a high quality daily global record of surface ocean temperature obtained from satellites, ships, buoys, and Argo floats on a regular grid. Its main input is infrared satellite data from the Advanced Very High Resolution Radiometer with high temporal-spatial coverage spanning late 1981 to the present. Any large-scale satellite biases relative to in-situ data from ships and buoys are corrected and any gaps are filled in by interpolation.

We use five different satellite-based estimates of NPP. The first is calculated by the NASA Ocean Biogeochemical Model (NOBM) (Gregg and Rousseaux, 2017; Gregg and Casey, 2007), a comprehensive ocean biogeochemical model coupled to a global ocean circulation and radiative model, which assimilates satellite ocean color data from the Sea-viewing Wide Field of View Sensor (SeaWiFs), the Moderate Resolution Imaging Specroradiometer (MODIS)-Aqua, and the Visible Infrared Imaging Radiometer Suite (VIIRS) to constrain NPP estimates over the mixed layer. The four other NPP datasets are based on the Vertically Generalized Production Model (VGPM) (Behrenfeld et al., 2005), that estimates NPP within the euphotic layer from chlorophyll or phytoplankton carbon concentrations, available light and a temperature-dependent description of photosynthetic efficiency. The four versions of this model are: Standard-VGPM, Eppley-VGPM, CbPM-VGPM and CAFE-VGPM (http://sites.science.oregonstate.edu/ocean.productivity/index.php). The only difference between Standard (Behrenfeld and Falkowski, 1997) and Eppley-VGPM is the temperature-dependent description of photosynthetic efficiencies: Standard-VGPM uses a polynomial function of temperature while Eppley-VGPM uses the exponential function described by Eppley (1972). Instead of deriving phytoplankton biomass from surface chlorophyll, the Carbon-based Production Model (CbPM; Behrenfeld et al. (2005); Westberry et al. (2008)) estimates phytoplankton carbon concentrations using coefficients of particulate scattering. And finally, CAFE-VGPM refers to the Carbon, Absorption, and Fluorescence Euphotic-resolving (CAFE) model (Silsbe et al., 2016), which calculates NPP as the product of energy absorption and the efficiency by which absorbed energy is converted into carbon biomass. VGPM-based models also use SeaWiFS, MODIS or VIIRS data. Fig. B1a-j in the Appendix provides the temporal mean and standard deviation of each observation-based NPP product. We chose to include all five observation-based NPP products as NPP estimates by models assimilating satellite data are still uncertain and highly sensitive to their respective model configurations (e.g. Behrenfeld et al. (2005); Long et al. (2021a)).

The SST and all satellite-derived NPP data used in this study are regridded to the coarser NOBM grid resolution of 1.25° longitude by 2/3° latitude for the period 1998 to 2018 before the analysis. The NOBM-based NPP product has a 5-day resolution, whereas the four VGPM-based NPP products have a 8-day resolution. From daily SST, we computed and used the 5-day mean SST when working with the 5-day mean NOBM-based NPP products, and the 8-day mean SST when working with



VGPM-based NPP. As NPP is close to or equal to zero during winter in the polar regions when solar radiation is near zero, we
follow the approach of Le Grix et al. (2021) and remove all days during which a particular grid cell receives no solar radiation,
thereby focusing on the growing season. The daily shortwave radiation data was obtained from the Modern-Era Retrospective
analysis for Research and Applications version 2 (Gelaro et al., 2017).

## 2.2  Model descriptions and large ensemble simulations

We use two global fully coupled Earth System Models (ESMs): GFDL's ESM2M and CESM2. ESM2M is a fully coupled
carbon–climate ESM developed at NOAA's Geophysical Fluid Dynamics Laboratory (GFDL) (Dunne et al., 2012, 2013). It
couples an atmospheric circulation model to an oceanic circulation model, and includes representations of land, sea ice, and
iceberg dynamics, as well as interactive biogeochemistry. The atmospheric model AM2 (Team et al., 2004) has a horizontal
resolution of $2°$ latitude x $2.5°$ longitude, and 24 vertical levels. The horizontal resolution of the ocean model MOM4p1
(Griffies, 2012) is nominally $1°$ latitude x $1°$ longitude with increasing meridional resolution of up to $1/3°$ towards the
equator, with 50 depth levels. Phytoplankton is represented in ESM2M by the biogeochemical module "Tracers of Ocean
Phytoplankton with Allometric Zooplankton version 2.0" (TOPAZv2; Dunne et al. (2013)), consisting of 30 tracers including
three phytoplankton groups (small and large phytoplankton, diazotrophs) and heterotrophic biomass (see section 2.4 for further
details). TOPAZv2 only implicitly simulates zooplankton activity. The large ensemble simulation ESM2M-LE was started from
a quasi-equilibrated 500-yr long preindustrial control simulation, where atmospheric $CO_2$ concentrations are set to 286 ppm
(Burger et al., 2020). We generated an ensemble of 15 members by slightly perturbing the temperature on the order of $10^{-5}$
$°C$ for five ensemble members at a grid cell at the surface of the Weddell Sea, for five members at the surface of the North
Atlantic and for five members in the deep North Pacific (Burger et al., 2021). These 15 simulations were forced with prescribed
historical concentrations of atmospheric $CO_2$ and non-$CO_2$ radiative forcing agents from 1861 to 2005, and then by following
a high-emission no mitigation scenario (RCP8.5; RCP: Representative Concentration Pathway) from 2006 to 2100 (Riahi et al.,
2011).

The Community Earth System Model version 2 (CESM2, Danabasoglu et al. (2020)) is also a fully coupled ESM. It couples
an atmospheric model with comprehensive chemistry to ocean, land, sea-ice, land-ice, river, and ocean wave models. The
horizontal resolution of the atmospheric model CAM6 (Danabasoglu et al., 2020) is $0.9°$ latitude x $1.25°$ longitude, with 32
vertical levels. The horizontal resolution of the ocean model POP2 (Smith and Gent, 2010) is approximately $1°$, with uniform
spacing of $1.125°$ in the zonal direction and varying significantly in the meridional direction, with the finest resolution of
$\sim0.25°$ at the Equator. The ocean model has 60 vertical levels. The "Marine Biogeochemistry Library" (MARBL; Long
et al. (2021b)) is the biogeochemical component of CESM2, which includes three phytoplankton types: small phytoplankton,
diatoms (i.e., large phytoplankton) and diazotrophs. It is a prognostic ocean biogeochemistry model that simulates marine
ecosystem interactions and the coupled cycles of carbon, nitrogen, phosphorus, iron, silicon, and oxygen. We use nine members
of a 100-member large ensemble simulation (CESM2-LE; (Rodgers et al., 2021)) in this study, for which all necessary 5-
day-mean data for the analysis was available. All members differ by their starting day, sampled at 20-year interval from a
preindustrial control simulation (Rodgers et al., 2021). Historical simulations were run from 1850 to 2014, forced by prescribed



atmospheric $CO_2$ concentrations and non-$CO_2$ radiative forcing agents. Projections from 2015 to 2100 follow the SSP3-7.0
scenario (Eyring et al., 2016).

ESM2M and CESM2 differ in their ecological module and how the latter computes phytoplankton growth and decay (see
Appendix A for a detailed description and comparison). For example in ESM2M, TOPAZv2 uses an Eppley function of temperature to represent the dependence of phytoplankton growth on temperature, whereas in CESM2, MARBL uses a power function following a $Q_{10}$ model (Sherman et al., 2016), resulting in weaker dependence of phytoplankton growth on temperature
in CESM2. Although both models represent the nutrient limitation on phytoplankton growth according to Michaelis-Menten kinetics, MARBL uses lower half-saturation constants for each nutrient than TOPAZv2. Assuming similar nutrient levels, this would imply lower nutrient limitation in CESM2. In addition to these differences, the ESM2M-LE is forced by RCP8.5 after 2006, whereas the CESM2-LE is forced by SSP3-7.0 after 2015. However, the different forcings applied do not impact our results, as the total radiative forcing of the two scenarios differ very little before year 2018 (Riahi et al., 2017), which is the
end point of our analysis period.

For both the ESM2M-LE and CESM2-LE, we select the historical period spanning from 1998 to 2018, over which we can compare the simulations to available satellite-derived observations of SST and NPP. Outputs are saved at 5-day mean resolution. They include SST, NPP, and all variables from which we analyze the drivers of NPP: phytoplankton biomass, growth, and loss terms (i.e., grazing in ESM2M, grazing, mortality and aggregation in CESM2), as well as the temperature,
light and nutrient limitations on phytoplankton growth. These variables are saved at a 10-meter vertical resolution. We integrate the phytoplankton NPP, biomass and loss terms over the upper 100-meter layer of the ocean, and compute biomass-weighted averages of phytoplankton growth and of its limitation terms by multiplying these variables with the biomass at each depth level, computing the vertical mean over the top 100 meters and dividing by the vertical mean biomass. Similarly as for the observation-based products, we focus on the growing season by removing all calendar days receiving no solar radiation (Gelaro
et al., 2017).

## 2.3 Definition of compound MHW-NPPX events

We subtract from each time series its mean seasonal cycle, which we smoothed to remove noise associated with the relatively short time series. For the observations, the smoothed seasonal cycle was obtained using a 30-day running average, and for ESM2M-LE and CESM2-LE, it was identified using their respective ensemble mean seasonal cycle. As we work with de-
seasonalized anomalies, compound events can occur throughout the year. At each grid cell, a MHW occurs when the SST anomaly exceeds its local 90th percentile. Respectively, an NPPX event occurs when the NPP anomaly is lower than its 10th percentile. A multivariate compound MHW-NPPX event occurs when both conditions are satisfied at the same time and location. Following this definition where no duration threshold is applied, extreme events can be as short as one time step, which here is a 5-day mean.

Due to their definition, univariate extreme events have the same frequency over the global ocean. At each grid cell, 10% of all time steps are MHWs and 10% are NPPX events. This implies that under the assumption of independence between MHW and NPPX events, the frequency of compound MHW-NPPX events would be 1% over the global ocean. Regions where





their frequency exceeds 1% can be considered hotspots of unusually frequent compound MHW-NPPX events. In our case, the frequency of compound events is equivalent to the likelihood multiplication factor, i.e., a measure of how many times more

frequent compound events are compared to their expected frequency under the assumption of independence (Zscheischler and Seneviratne, 2017; Le Grix et al., 2021; Woolway et al., 2021; Burger et al., 2021).

## 2.4    Model evaluation

The Taylor diagrams presented in Fig. 1 provide a summary of the relative skill with which the models simulate the mean and variability of SST and NPP as well as the extreme event magnitude (i.e., mean SST and NPP anomalies during extreme

events relative to their climatological mean values) and duration of MHWs and NPPX events. The simulated patterns of the mean state of SST by ESM2M-LE and CESM2-LE are very similar to that computed from the observation-based SST (r>0.99 and normalized std $\sim$ 1, red point and cross in Fig. 1a). The CESM2-LE is slightly better than ESM2M-LE at simulating the pattern of temporal variability of SST (r=0.8 for ESM2M-LE and r=0.9 for CESM2-LE, Fig. 1b). The globally integrated NPP is 74 Pg C year$^{-1}$ in ESM2M-LE and 43 Pg C year$^{-1}$ in CESM2-LE, compared to 53 Pg C year$^{-1}$ on average (range of 46 to

62 Pg C year$^{-1}$) in the observation-based estimates (Fig. B1). ESM2M-LE substantially overestimates NPP, especially in the low latitudes where the simulated NPP exceeds 1000 mg C m$^{-2}$ day$^{-1}$ compared to the observation-based estimate of about 400-800 mg C m$^{-2}$ day$^{-1}$. Despite these differences, ESM2M-LE and CESM2-LE succeed in representing the NPP mean spatial pattern of higher values in the low latitudes and lower values in the subtropical gyres and in the Southern Ocean. These results are summarized on Fig. 1a, where the different observation-based NPP products are as dispersed as ESM2M-LE and

CESM2-LE themselves, indicating that the models are approximately within the range of the observations. The NPP temporal variability simulated by the two models is also similar to that estimated by the observation-based products (Fig. 1b, Fig. B1, right column), although the models underestimate the spatial heterogeneity in the NPP temporal variability pattern (normalized std<0.25).



**Figure 1.** Comparative assessment of the simulated mean and extreme states of SST and NPP to an observed reference. These Taylor diagrams compare the spatial pattern of the climatological mean state (a) and standard deviation (b) of 5-day mean SST and NPP, as well as of the magnitude (c) and 90th percentile of the duration (d) of MHWs and NPPX events, simulated by each model to that of a reference. The reference is calculated from the observation-based SST estimate or from the mean of the five different observation-based NPP estimates, and it is indicated by a star on the diagrams. A circle, a triangle and the numbers 1, 2, 3, 4 and 5 represent ESM2M-LE, CESM2, NOBM, VGPM-Standard, VGPM-Eppley, VGPM-CbPM and VGPM-CAFE, respectively. The Pearson correlation coefficient, which quantifies similarity between the simulated pattern and the reference, is indicated by the azimuthal angle; the centered RMS error in the simulated field is proportional to the distance from the star on the x-axis; and the standard deviation of the simulated pattern is indicated by the radial distance from the origin. All statistics are normalized by the standard deviation of the reference.



The MHW magnitudes identified from the satellite-based observations are similar to those simulated by ESM2M-LE and

CESM2-LE (Fig. 1c, Fig. B2a-c). However, both models simulate MHWs that last longer than those in the observations (Fig. B2d-f), especially in the eastern equatorial Pacific. This is a common bias across all current global Earth system models (Frölicher et al., 2018), irrespective of their vertical and horizontal resolution (Pilo et al., 2019). The spatial pattern of MHW duration is reasonably well simulated in both models (Fig. 1d). In contrast, the models differ in their representation of NPPX events (Fig. 1c-d, Fig. B3). The observation-based mean NPPX magnitude is most intense ($< 250$ mg C m$^{-2}$ day$^{-1}$) in

the tropical Atlantic Ocean and in the northern high latitudes, whereas the magnitude is most intense in ESM2M-LE in the equatorial Pacific and in the Indian Ocean, and in CESM2-LE in the northern high latitudes and in the Southern Ocean. Given the low agreement between the observation-based NPP products (Fig. 1c), it is difficult to assess how well ESM2M-LE and CESM2-LE simulate the magnitude of NPPX events, and which of the two models is more realistic. We also compare the 90th percentile of the duration of NPPX events (Fig. 1d) to highlight differences between the observations and the models

even though their observed and simulated median duration is close to 5 days over the global ocean due to the predominance of short NPPX events. In the observations, NPPX events reach their longest durations ($> 70$ days) in the central equatorial Pacific (Fig. B3d). The spatial patterns simulated by the models for the NPPX events duration differ from that of the observed reference (r$<$0.2 for ESM2M-LE and for CESM2-LE, Fig. 1d). In ESM2M-LE, the longest events ($> 90$ days) occur within the subtropical gyres, where NPP anomalies do not vary much over time (Fig. B3e, normalized and centered RMS error $= 4.3$

on Fig. 1d). In CESM2-LE, events are of short duration over most of the global ocean and slightly longer ($> 30$ days) in the high latitudes and in the eastern equatorial Pacific (Fig. B3f).

Overall, ESM2M-LE and CESM2-LE represent the mean state and variability of SST and NPP reasonably well and appear therefore suited to investigate the likelihood and drivers of compound MHW-NPPX events over the global ocean. However, divergent magnitudes and durations of NPPX events in ESM2M-LE and CESM2-LE hint at different drivers for NPPX events

in the two models. Different processes might thus drive NPPX in association with a MHW and result in a compound MHW-NPPX event in ESM2M-LE and CESM2-LE.

## 2.5 Driver decomposition of compound MHW-NPPX events

We investigate the drivers of compound MHW-NPPX events, and more specifically the drivers of extreme reductions in NPP during MHWs. Both ESM2M and CESM2 contain an ecological module distinguishing between three different phytoplankton

types (small, large and diazotrophs), for which specific constants and limitation terms are used to compute NPP. Total net phytoplankton production is simply the sum of NPP over all three phytoplankton types. Thus, during a low NPP event, although the total phytoplankton NPP is extremely low, not all types may have contributed to that anomaly. We ignore the diazotrophs in this study as their contribution to total NPP ($1.5\%$ in ESM2M-LE and $3\%$ in CESM2-LE on average) and to the total NPP anomaly during compound MHW-NPPX events ($< 0.1\%$ in ESM2M-LE and CESM2-LE) is negligible. Thus, in each model,

total NPP is approximated as the sum of small and large phytoplankton NPP.





For each phytoplankton type $i$, NPP is the product of its growth rate $\mu$ and its biomass $n$:

$$NPP_i = \mu_i n_i \tag{1}$$

Therefore, any anomaly in NPP, $dNPP$, can be decomposed as:

$$dNPP_i = n_i d\mu_i + \mu_i dn_i \tag{2}$$

If $dNPP$ stands for the mean NPP anomaly during compound events relative to the (smoothed) climatological mean state of the seasonal cycle, we can assess the contributions of the mean growth anomaly, $d\mu$ and of the mean biomass anomaly, $dn$, during compound events to $dNPP$.

    TOPAZv2 and MARBL define $\mu$ in the same way:

$$\mu_i = \mu_{max_i} T_{f_i} L_{lim_i} N_{lim_i} \tag{3}$$

where $T_f$ is a function of the temperature, $L_{lim}$ is the light limitation, and the nutrient limitation $N_{lim}$ is computed using Leibig's law of the minimum. More details are provided in the Appendix A. Both $N_{lim}$ and $L_{lim}$ are between 0 and 1, where 1 means they do not limit phytoplankton growth and 0 means they fully suppress growth. Fig. B4 in the Appendix presents the climatological mean states of $T_f$, $L_{lim}$ and $N_{lim}$.

    $d\mu$ can be decomposed into the contributions of a change in $T_f$, $L_{lim}$, and $N_{lim}$ during compound events.

$$d\mu_i = \mu_{max_i}(N_{lim_i}L_{lim_i}dT_{f_i} + N_{lim_i}T_{f_i}dL_{lim_i} + T_{f_i}L_{lim_i}dN_{lim_i}) \tag{4}$$

    This decomposition enables us to assess the drivers of a change in phytoplankton growth during compound events. Drivers of a change in phytoplankton biomass are less trivial, as $n$ depends on NPP itself. In TOPAZv2 and MARBL, $n$ is considered a tracer whose time derivative is defined as:

$$\partial_t n_i = NPP_i - Loss_i - \nabla(\overrightarrow{u}.n_i) \tag{5}$$

where $NPP - Loss$ are its biological production and decay, respectively, and $-\nabla(\overrightarrow{u}.n)$ corresponds to the physical advection and mixing of phytoplankton biomass by ocean circulation. Any negative $dn$ contribution to $dNPP$ is driven by a decrease in biomass over time. To assess the contributions of biology and circulation to $dn$, we integrate $\partial_t n$, $NPP$ and $Loss$ over all periods over which $dn$ builds up, i.e., over which the biomass changes from its climatological mean value to its maximum absolute value reached during a compound event. $\Delta n$ refers to the integrated biomass change $\int \partial_t n dt$. From the difference

between $\Delta n$ and its $NPP - Loss$ contribution, we infer the circulation contribution. Details on the computation of phytoplankton $Loss$ are provided in Sections A1.5 for TOPAZv2 and A2.5 for MARBL, and Fig. B5 presents the climatological mean states of $NPP$, $Loss$, $n$ and $\mu$.





## 3 Results

### 3.1 Hotspots of compound MHW-NPPX events in the global ocean

Figure 2 shows the present-day distribution of compound MHW-NPPX events. Under the assumption of independence between MHW and NPPX events, the expected frequency of compound MHW-NPPX events is 1% of time intervals or 3.65 days/year over the global ocean (Section 2.3). However, observation-based estimates show strong regional deviations from this expected frequency (Fig. 2a). Most compound events occur in the low latitudes, with hotspots of especially high frequency in the center of the equatorial Pacific, the subtropical Indian Ocean and around Antarctica. In these regions, compound MHW-NPPX events

occur more than 3 times more frequently ($> 3\%$ or $> 10$ days/year) than would be expected if univariate extremes were independent. Compound MHW-NPXX events are also relatively frequent (about 2% or 7 days/year) in the low to mid latitudes between $10°$ and $45°$. In contrast, compound events are rare (about 0.5% or 2 days/year) in the high northern latitudes north of 45°N and between 45°S and 60°S in the Southern Ocean. However, these estimates correspond to the mean of the results obtained from five observation-based NPP products, which disagree particularly in the high southern latitudes and somewhat in

the low latitudes (Fig. 2d; Fig. B6). Around Antarctica, the frequency computed using NOBM's NPP is much lower on average (0.5%) than those computed using VGPM-based NPP products ($> 4\%$). Sea-ice and clouds create gaps in the satellite ocean color data that are potentially more extended in time and space around Antarctica than over the rest of the global ocean. Sparse satellite data coverage implies that in NOBM, fewer ocean color observations are available to constrain NPP estimates, whereas in VGPM-based models, gaps are filled by interpolation with data points that might be too distant in space and time to yield a

realistic estimate of NPP (Rousseaux and Gregg (2014); http://orca.science.oregonstate.edu/gap_fill.php). For this reason, we have lower confidence in the NOBM and VGPM-based NPP products around Antarctica than elsewhere. In the low to mid latitudes, the frequency computed using Standard-VGPM is higher than that of all other observation-based estimates (Fig. 2d). Standard VGPM is the only model that uses a polynomial function to describe the temperature dependence of photosynthesis. Therefore, extremely hot surface waters in the warm low to mid latitudes have a weaker positive effect on photosynthesis

and thus on NPP in Standard-VGPM than in the other observation-based products. It may thereby be easier for high SST to co-occur with low NPP, resulting in higher frequency of compound MHW-NPPX events in Standard-VGPM in the low to mid latitudes.





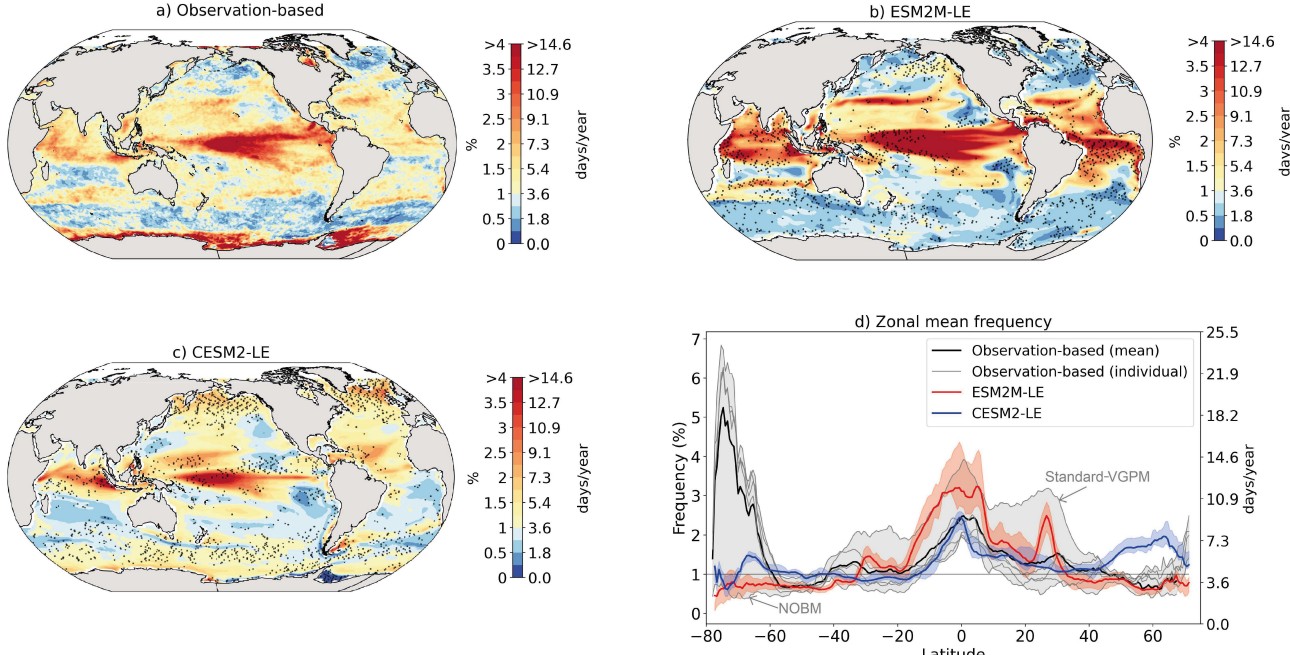

**Figure 2.** Frequency of compound MHW-NPPX events in (a) observation-based estimates, and as simulated by (b) ESM2M-LE, and (c) CESM2-LE. Observations correspond to the mean of the results obtained with 5 satellite-based estimates of NPP, namely NOBM, Standard-VGPM, Eppley-VGPM, CbPM-VGPM, and CAFE-VGPM. (d) Zonal mean frequency of compound MHW-NPPX events. The grey, red and blue shaded areas in (d) indicate the range of the observation-based estimates, of the ESM2M-LE members and of the CESM2-LE members, respectively. Stipplings in (b) and (c) correspond to regions where the frequency simulated by ESM2M-LE and CESM2-LE is outside the range of the observation-based estimates, i.e., higher or lower than all 5 observation-based estimates.

Next, we compare the simulated frequency of compound MHW-NPPX events in ESM2M-LE (Fig. 2b) and CESM2-LE (Fig. 2c) to the observation-based frequency (Fig. 2a,d). Overall, the simulated frequency pattern is similar in the two models and mostly within the uncertainty range of the observational products (e.g., areas with no stippling in Fig. 2b and Fig. 2c,
corresponding to 84% of the global ocean in ESM2M-LE and to 82% in CESM2-LE). The models correctly simulate frequent compound MHW-NPPX events in the equatorial Pacific ($> 4\%$ or $> 14$ days/year) and relatively frequent compound events in the low to mid latitudes between $10°$ and $45°$ ($2\%$ or $7$ days/year; Fig. 2a-c). ESM2M-LE simulates too frequent compound events in the southern tropical Atlantic, in the center of the equatorial Pacific and in the northern part of Indian Ocean. CESM2-
LE simulates too frequent compound events in the western equatorial Pacific and in the northern part of Indian Ocean. In spite of there being relatively few dissimilarities between models and observations in the low and mid latitudes, they strongly disagree in the high latitudes. ESM2M-LE slightly outperforms CESM2-LE, especially in the northern high latitudes, where it simulates rare compound events consistent with the observation-based estimates, whereas CESM2-LE simulates too frequent compound events ($> 1\%$). Around Antarctica, neither ESM2M-LE nor CESM2-LE simulate the very frequent compound MHW-NPPX
events shown in the observations. However, low agreement between the five observation-based estimates (their frequency being





as low as $0.5\%$ and as high as $6.5\%$ on average at 75°S, in Fig. 2d) makes it difficult to determine which of the two models better simulates compound events in this regions.

## 3.2 Small and large phytoplankton NPP anomalies during compound MHW-NPPX events

Next, we assess which phytoplankton type is responsible for NPPX during compound MHW-NPPX events. In both models,
total NPP is approximately equal to the sum of small and large phytoplankton NPP (Section 2.5), whose respective mean anomalies (relative to the mean seasonal cycle) during compound MHW-NPPX events is presented in Figure 3a-d.

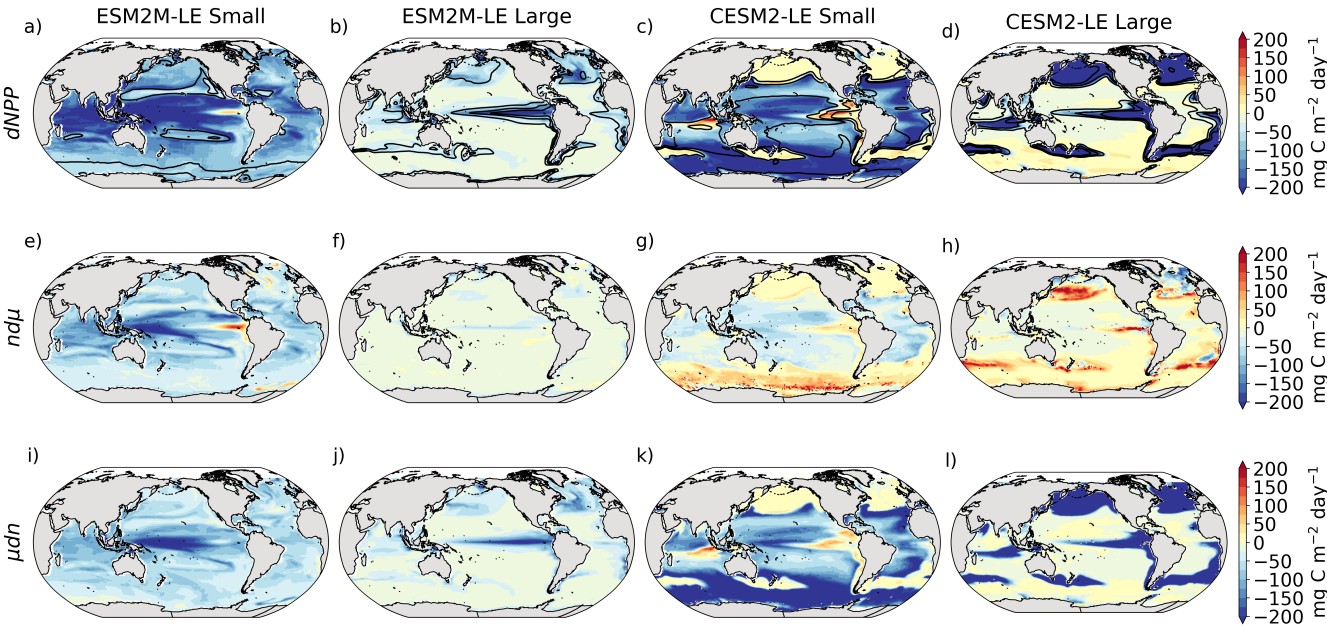

**Figure 3.** Small and large phytoplankton NPP anomalies $dNPP$ relative to the climatological seasonal cycle (mg C m$^{-2}$ day$^{-1}$) during compound MHW-NPPX events in ESM2M-LE (a,b) and in CESM2-LE (c,d), and contributions of the growth rate $nd\mu$ (e-h) and of the biomass anomaly $\mu dn$ (i-l) to these NPP anomalies. Contours on panels a-d indicate the climatological mean state of small and large NPP averaged over 1998-2018 (see also Fig. B5a-d); labels have been omitted.

The decrease in small phytoplankton NPP dominates the overall decrease in NPP during compound MHW-NPPX events in both models, although the models differ in the magnitude and spatial pattern of anomalies in small and large phytoplankton. The decrease in small phytoplankton NPP accounts for $79\%$ and $70\%$ of the total NPPX anomalies in the global ocean during
MHW-NPPX events in ESM2M-LE and CESM2-LE, respectively (Fig. 3a,c). Especially pronounced is the dominance of small phytoplankton NPP decreases in the low to mid latitudes and the Southern Ocean in both models. This implies a shift in the phytoplankton community composition from small phytoplankton towards relatively more large phytoplankton during MHW-NPPX events in these regions, with potential repercussions for marine communities structure. In both models, decreases in large phytoplankton NPP dominate the NPP decrease during MHW-NPPX events in the eastern equatorial Pacific. Large





phytoplankton NPP also decreases during MHW-NPPX events in the northern high latitudes. In CESM2-LE, the decline in
large phytoplankton NPP even dominates the response in the northern high latitudes as small phytoplankton NPP increases,
resulting in an assemblage shift towards smaller phytoplankton there. In addition, the decline in large phytoplankton NPP also
dominates along the southern boundaries of the subtropical gyres in the Southern Hemisphere in CESM2-LE. Overall, these
patterns resemble well the climatological mean state pattern of small and large phytoplankton NPP (see contour lines in Fig. 3a-

d and Fig. B5a-d). Small phytoplankton anomalies during MHW-NPPX events dominate in regions where the climatological
mean state of small phytoplankton NPP generally dominates, whereas large phytoplankton NPP anomalies play an important
role during MHW-NPXX events where the abundance of large phytoplankton is relatively high in the climatological mean.

### 3.3 Drivers of low NPP during compound MHW-NPPX events

To understand the drivers of NPPX during compound MHW-NPPX events, we decompose the NPP anomaly $dNPP$ of each
phytoplankton type into the contributions of its growth rate anomaly $d\mu$ (Fig. 3e-h) and of its biomass anomaly $dn$ (Fig. 3i-l)
(see equation 2 in Section 2.5). One must note, however, that these variables are not independent and that the biomass anomaly
may result from changes in the growth rate itself. The decomposition amounts to 104% and 105% of the global $dNPP$ of
small and large phytoplankton in ESM2M-LE, and to 104% and 99% of the global $dNPP$ of small and large phytoplankton
in CESM2-LE, respectively (Fig. B7). Our decomposition method is therefore well suited to investigate the drivers of extreme
reductions in NPP during MHX-NPPX events.

Globally, the growth rate anomaly $d\mu$ barely contributes to the large phytoplankton $dNPP$ in ESM2M-LE (28%, Fig. 3b,f)
and to the small and large phytoplankton $dNPP$ in CESM2-LE ($-12\%$ and $-14\%$, respectively Fig. 3c,d,g,h). A large part
of the extreme reduction in NPP during MHWs is in fact driven by a negative biomass anomaly $dn$ in both models and for
both phytoplankton types. However, the growth rate anomaly explains about half (51%) of the global small phytoplankton
$dNPP$ in ESM2M-LE (Fig. 3a,e) and can regionally be even more dominant. In ESM2M-LE, the contribution of $d\mu$ is most
negative in the low latitudes for small phytoplankton (Fig. 3e), especially in the western equatorial Pacific. In CESM2-LE, the
contribution of $d\mu$ is slightly negative in the low latitudes (Fig. 3g), and positive (i.e., it counteracts the negative $dNPP$) in the
high latitudes and eastern equatorial Pacific for small and large phytoplankton (Fig. 3g,h). In other words, an increase in the
growth rate increases small and large phytoplankton NPP in these regions in CESM2-LE. However, the large decreases in $dn$
overcompensate this increase in growth rate and leads to an overall decrease of NPP for small phytoplankton in the low to mid
latitudes and in the high southern latitudes (Fig. 3k), and for large phytoplankton in the eastern equatorial Pacific, in the high
northern latitudes and at around 40°S (Fig. 3l).

Increases in small or large phytoplankton NPP moderate the negative $dNPP$ during MHW-NPPX events. In ESM2M-
LE, small phytoplankton NPP locally increases in the eastern equatorial Pacific as a result of increased small phytoplankton
growth (Fig. 3e). In CESM2-LE, the increase in small phytoplankton NPP in the northern high latitudes and the increase in
large phytoplankton NPP in the southern high latitudes are driven by both an increase in growth and an increase in biomass
(Fig. 3g,h,k,l).





### 3.3.1 Phytoplankton growth rate anomaly during compound MHW-NPPX events

Before explaining the changes in phytoplankton biomass, we look into the drivers of changes in phytoplankton growth rates, as
they contribute to reducing NPP either directly, or indirectly by affecting phytoplankton biomass. Figure 4 shows the spatial
pattern of the mean growth rate anomaly $d\mu$ during compound events for small and large phytoplankton in each model, as well
as the contributions of temperature, light and nutrient limitations to $d\mu$, as described in Section 2.5.

In ESM2M-LE, the drivers of $d\mu$ are similar during compound events for small and large phytoplankton. The negative growth
rate anomaly in the low to mid latitudes (Fig. 4a,b) is associated with increased nutrient limitation (-0.10 day$^{-1}$ on average
between 40°S and 35°N; Fig. 4m,n), i.e., reduced mixing of nutrient-rich waters from the deeper ocean to the upper 100m.
In the high latitudes, the negative growth rate anomaly is mainly associated with increased light limitation (-0.05 day$^{-1}$ on
average south of 40°S and north of 35°N; Fig. 4i,j). High temperatures during MHWs somewhat moderate the negative growth
rate anomalies by their positive effect on the growth rate for both large and small phytoplankton, especially in the low latitudes
(Fig. 4e,f). In the eastern equatorial Pacific, this positive effect of the temperature is able to overcompensate for the negative
effect of nutrient limitation on the growth rate of small phytoplankton (Fig. 4e), resulting in increased small phytoplankton
growth and a shift towards small phytoplankton during MHW-NPPX events (Fig. 3a,b).

In CESM2-LE, $d\mu$ is negative in the low latitudes (Fig. 4c,d) for both small and large phytoplankton. The growth of small
phytoplankton is mainly reduced by increased nutrient limitation (-0.05 day$^{-1}$ on average between 15°S and 20°N (Fig. 4o,p),
whereas the growth of large phytoplankton is mainly reduced by light limitation in the low latitudes (-0.03 day$^{-1}$ on average
between 20°S and 20°N; Fig. 4l). In the high latitudes, small and large phytoplankton growth is enhanced due to reduced light
limitation. High temperature anomalies contribute positively to the growth rate of small and large phytoplankton, especially in
the eastern equatorial Pacific for small phytoplankton (>0.09 day$^{-1}$; Fig. 4g), resulting in a shift towards large phytoplankton
during MHW-NPPX events there (Fig. 3c,d).




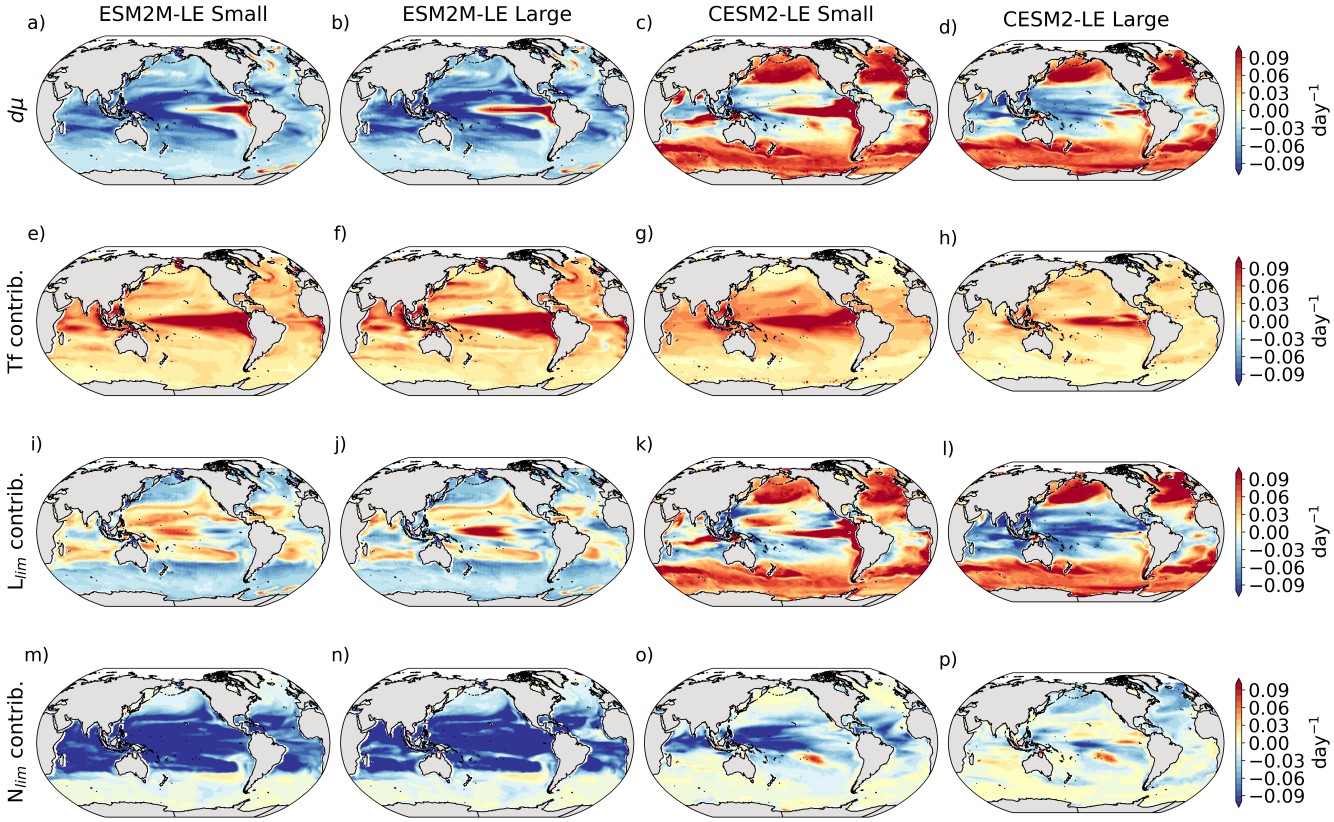

**Figure 4.** Growth rate anomaly (day$^{-1}$) of small and large phytoplankton during compound MHW-NPPX events in ESM2M-LE (a,b) and in CESM2-LE (c,d), and contributions of a change in the temperature function (e-h), in the light limitation (i-l) and in the nutrient limitation (m-p) to this growth rate anomaly.

Overall, the models agree that phytoplankton growth is enhanced by high temperatures and reduced by low nutrient levels during MHW-NPPX events. However, the models disagree on the strength of the nutrient limitation changes, potentially due to TOPAZv2 in ESM2M-LE using higher half-saturation constants than MARBL in CESM2-LE. The models also disagree on their representation of the light limitation changes during MHW-NPPX events, especially in the high latitudes. In ESM2M-LE, MHW-NPPX events are associated with enhanced light limitation on phytoplankton growth in the high latitudes, whereas in CESM2-LE, they are associated with reduced light limitation in the high latitudes, which, along with the positive temperature effect, enhances the growth rate and explains the positive $d\mu$ in the high latitudes. This model divergence may arise from a number of factors involved in the calculation of $L_{lim}$, such as different nutrient limitations, chlorophyll to carbon ratios and light harvest coefficients for small and large phytoplankton in TOPAZv2 (Section A1.3) and MARBL (Section A2.3), resulting in a different response of $L_{lim}$ to environmental changes during MHW-NPPX events.





### 3.3.2 Phytoplankton biomass anomaly during compound MHW-NPPX events

Next, we investigate the drivers of the mean phytoplankton biomass anomaly $dn$ during compound MHW-NPX events (Fig. 5a-d), which contributes to driving $dNPP$. The spatial pattern of $dn$ resembles the spatial pattern of $dNPP$ (Fig. 3a-d); their Pearson's correlation coefficient is 0.4, 0.9 for small and large phytoplankton in ESM2M, and 0.8 and 0.9 for small and large phytoplankton in CESM2, respectively. In ESM2M-LE, the negative $dn$ is rather uniform across latitudes for small phytoplankton (Fig. 5a), but shows a distinct spatial pattern for large phytoplankton with stronger declines in the eastern

equatorial Pacific and in the high northern latitudes (Fig. 5b). In CESM2-LE, low NPP is driven by a decrease in small phytoplankton biomass in the southern high latitudes and partly in the low latitude (Fig. 5c), and by a decrease in large phytoplankton biomass along the equator, in the northern high latitudes and in the southern boundary of the subtropical gyres of the Southern Hemisphere (Fig. 5d).

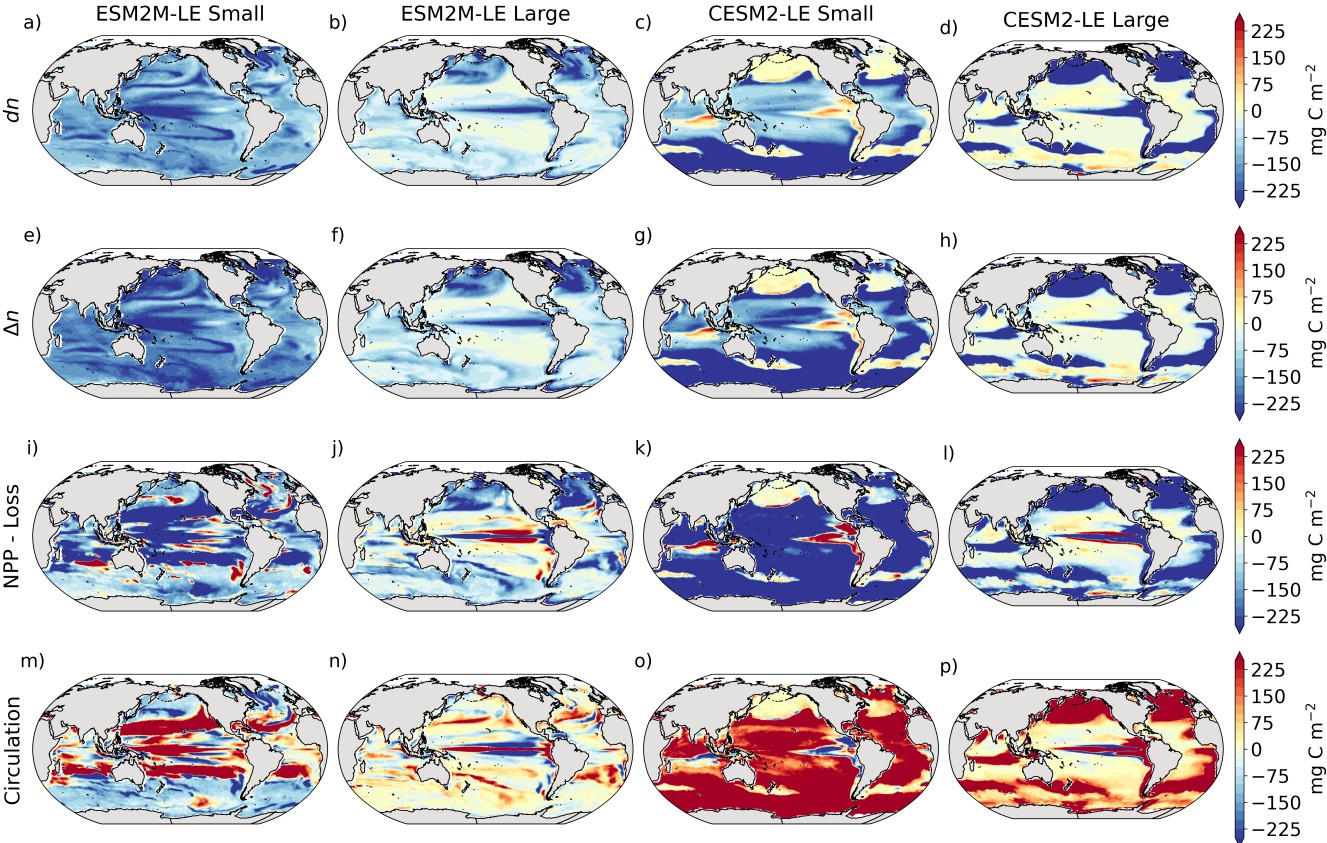

**Figure 5.** Biomass anomaly $dn$ (mg C m$^{-2}$) of small and large phytoplankton during compound MHW-NPPX events in ESM2M-LE (a,b) and in CESM2-LE (c,d). Integrated biomass change $\Delta n$ (mg C m$^{-2}$ day$^{-1}$) leading to the maximum anomaly reached during a compound MHW-NPPX event (e-h) and contributions (mg C m$^{-2}$ day$^{-1}$) of biology ($NPP - Loss$, i-l) and circulation ($\nabla(\overrightarrow{u}.n_i)$, m-p) to $\Delta n$.



We are further interested in the buildup of this biomass anomaly $dn$ over time. $\Delta n$ is the integrated biomass change over the
period over which biomass anomalies build up (Section 2.5). Even though $dn$ and $\Delta n$ differ by definition, they have almost
identical spatial patterns (Fig. 5a-d compared to Fig. 5e-h), signifying it is indeed possible to understand $dn$ from $\Delta n$.

$\Delta n$ is driven by changes in the difference between phytoplankton $NPP$ and $Loss$ ($NPP$ - $Loss$; Fig. 5i-l), and by changes
in ocean circulation (Fig. 5m-p) (see Equation 5). Although the individually integrated $NPP$ and $Loss$ terms seem almost
equivalent (Fig. B8), phytoplankton loss actually exceeds phytoplankton NPP over most of the global ocean (Fig. 5i-l), which
contributes to decreasing the biomass over time (Fig. 5e-h), and thus to driving the negative biomass anomaly $dn$ (Fig. 5a-d). In
ESM2M-LE, this effect is particularly strong ($< -150 \, \mathrm{mg \, C \, m^{-2}}$) for small phytoplankton in the low to mid latitudes between
35°S and 35°N, and for large phytoplankton in the northern high latitudes and in a narrow band along the equator (Fig. 5i,j). In
CESM2-LE, the negative $NPP$ - $Loss$ contribution to $\Delta n$ is especially strong ($< -200 \, \mathrm{mg \, C \, m^{-2}}$) for small phytoplankton
in the low to mid latitudes between 35°S and 35°N and in the Southern Ocean, and for large phytoplankton along the Equator
and in the high latitudes (Fig. 5k,l).

Ocean circulation generally moderates the biomass decrease driven by $NPP$ - $Loss$ by importing more or exporting less
phytoplankton in or out of the top 100-meter layer of the ocean (Fig. 5m-p). During MHWs, reduced downward mixing due
to stronger upper ocean stratification might, for example, retain phytoplankton in the surface layers of the ocean. Despite the
predominantly positive contribution of circulation to $\Delta n$, there are exceptions in ESM2M-LE, where circulation contributes
to decreasing the biomass of small phytoplankton in the high latitudes and of large phytoplankton over a limited area in the
eastern equatorial Pacific. In CESM2-LE, although the circulation contribution is nowhere negative enough to drive a negative
biomass anomaly, it moderates the positive biomass anomaly of small phytoplankton in the eastern equatorial Pacific.

Overall in both models, the negative biomass anomaly $dn$ (Fig. 5a-d) can be explained by negative biomass changes $\Delta n$
(Fig. 5e-h) over time, mostly driven by negative contributions from $NPP$ - $Loss$ (Fig. 5m-p). Loss terms include grazing in
TOPAZv2, and grazing, mortality and aggregation in MARBL. During MHWs, higher temperatures not only enhance NPP via
their positive effect on the growth rate, they also enhance phytoplankton loss via their similarly positive effect on phytoplankton
grazing and mortality (see Sections A1.5 and A2.5). However, other factors such as nutrient and light limitation moderate phy-
toplankton growth during compound MHW-NPPX events, as we have seen in the previous section. In turn, nutrient and/or light
limitation might moderate NPP sufficiently for it to be exceeded by phytoplankton loss, allowing a decrease in phytoplankton
biomass over time.

### 3.3.3 Summary of driving processes

Figure 6 summarises the drivers of NPPX during MHWs in ESM2M-LE and in CESM2-LE. We distinguish between four
regions of rather homogeneous drivers: the northern high latitudes north of 35°N, the low latitudes between 35°S and 35°N,
except for the eastern equatorial Pacific (as defined by Fay and Mckinley (2014)), and lastly the southern high latitudes south
of 35°S. Small and large phytoplankton contributions to $dNPP$ are represented on Fig. 6 by dark and light colors, respectively.
Here, we compare the drivers of NPPX in the two models and choose to not focus on the magnitude of their NPP anomalies
(note the different y-axis in Fig. 6). Small and large phytoplankton both contribute to driving NPPX during compound MHW-





NPPX events. In the two models, small phytoplankton is responsible for the majority ($> 70\%$) of $dNPP$ in the low latitudes and in the southern high latitudes. In ESM2M, large phytoplankton accounts for a larger part ($44\%$) of $dNPP$ in the northern high latitudes and about half of $dNPP$ over the cold tongue, whereas in CESM2-LE, large phytoplankton dominates ($> 84\%$) $dNPP$ in the northern high latitudes and over the cold tongue.







**Figure 6.** Regional mean NPP anomaly (mg C m$^{-2}$ day$^{-1}$) during compound MHW-NPPX events in ESM2M-LE and in CESM2-LE over the northern latitudes (a,b), the low latitudes (c,d), the eastern equatorial region (e,f) and the southern latitudes (g,h). Contributions of the small and large phytoplankton $dNPP$ to the total NPP anomaly are represented in black and gray, respectively. The indirect contributions to $dNPP$ of a change in the temperature function Tf (in red), in the light limitation $L_{lim}$ (in orange), in the nutrient limitation $N_{lim}$ (in blue), in $NPP - Loss$ (in green) and in circulation (in purple) during compound MHW-NPPX events are indicated in dark and light colors for small and large phytoplankton, respectively.





We further decomposed $dNPP$ into the contributions from a change in the temperature function $Tf$ (red bars in Fig. 6), in the light limitation $L_{lim}$ (yellow bars), and in the nutrient limitation $N_{lim}$ (blue bars) by multiplying their contributions to the growth rate anomaly $d\mu$ (Section 3.3.1) with the climatological mean biomass $n$. We also assessed the contributions

of a change in $NPP - Loss$ (green bars) and in circulation to $dNPP$ (purple bars) by multiplying their contributions to the biomass anomaly $dn$ (Section 3.3.2) with the climatological mean growth rate $\mu$. Over all four regions and in both models, high temperatures during MHWs have a positive effect on the growth rate and thus positively contribute to $dNPP$. This positive effect can be supported or counteracted by the light and nutrient contributions to $dNPP$. The models agree on the sign of the light contribution in the low latitudes: enhanced light levels increase phytoplankton growth and thus moderate the negative

NPP anomaly during MHW-NPPX events. Nevertheless, the models disagree in the high latitudes and in the equatorial Pacific. Although in CESM2-LE, enhanced light levels during MHW-NPPX events have for the most part a positive effect on $dNPP$ except on large phytoplankton in the equatorial Pacific (Fig. 6b,f,h), in ESM2M-LE, strong light limitation on phytoplankton growth contributes to reducing $dNPP$ and thus to driving NPPX events in the high latitudes (Fig. 6a,g).

The models agree that lower nutrient levels limit phytoplankton growth during compound MHW-NPPX events. However,

they disagree on the strength of the nutrient limitation, potentially due to TOPAZv2 using higher half-saturation constants than MARBL. In ESM2M-LE, the nutrient limitation on phytoplankton growth is strong enough (in combination with the light limitation in the high latitudes) to reduce the growth rate, which directly contributes to reducing NPP and thus to driving NPPX events (Fig. 6a,c,e,g). On the other hand, in CESM2-LE, the nutrient limitation is not sufficient to counterbalance the positive effects of temperature and light on the growth rate during MHWs in the high latitudes and over the cold tongue (Fig. 6b,d,h),

and only slightly contributes to reducing $dNPP$ in the low latitudes (Fig. 6f). No matter how strong the nutrient and/or light limitations are, they moderate and sometimes overcompensate the positive effect of temperature on NPP during MHWs. High temperatures also enhance phytoplankton $Loss$. The models agree that phytoplankton loss exceeds NPP over all four regions during MHW-NPPX events. This $NPP - Loss$ contribution causes the phytoplankton biomass to decrease over time and to build a negative anomaly, which contributes to driving NPPX events by about 50% in ESM2M-LE and $> 100\%$ in CESM2-

LE. This effect is generally counteracted by ocean circulation which tends to increase the biomass over time, possibly because MHWs are associated with stratification and reduced export of phytoplankton to the deeper ocean. We note that circulation exceptionally reduces $dNPP$ of small phytoplankton in the Southern Ocean in ESM2M-LE.

Overall, the models agree on the effect of high temperatures, which tend to increase NPP. They agree on higher light levels, which also increase NPP except in the high latitudes for ESM2M where light limitation contributes to driving NPPX. And

they finally agree on nutrient limitation during compound MHW-NPPX events which contributes to driving NPPX. The main difference between ESM2M-LE and CESM2-LE is their respective strength of the nutrient limitation effect on phytoplankton growth during MHW-NPPX events. In ESM2M-LE, the nutrient limitation is strong enough to reduce the growth rate and directly drive NPPX. In CESM-LE, weaker nutrient limitation simply moderates the temperature effect on the growth rate and thus on NPP, thereby potentially allowing NPP to be exceeded by phytoplankton loss, which decreases the biomass over time

and eventually drives NPPX.



## 4 Discussion and Conclusion

We had three primary goals in setting out with this study: (i) identify hotspots of compound marine heatwave and low NPP (MHW-NPPX) events, (ii) assess the fidelity of state-of-the-art Earth system models (ESMs) in representing MHW-NPPX events, and (iii) apply the models to develop mechanistic insights into the underlying drivers of these potentially harmful
compound MHW-NPPX events.

The analysis revealed that compound MHW-NPPX events occur relatively frequently in the low latitudes, especially in the center of the equatorial Pacific and in the subtropical Indian Ocean, and less frequently in the northern high latitudes (Figure 2a,d; first goal). Both models agree with observations in the low latitudes (second goal). However, CESM2-LE overestimates the frequency of compound MHW-NPPX events in the northern high latitudes. In the southern high latitudes, elevated
uncertainty in the observation-based products renders it difficult to determine which of the two models better simulates compound events. Overall, our results agree with previous studies that reported suppressed NPP during MHWs in regions with relatively low surface nutrient levels, such as the subtropical gyres (Hayashida et al., 2020; Gupta et al., 2020; Le Grix et al., 2021). Gupta et al. (2020), for example, reported low chlorophyll during a MHW in the Indian Ocean, where background nitrate concentrations are especially low. Le Grix et al. (2021) described frequent co-occurrence of MHWs and low chloro-
phyll events in the center of the equatorial Pacific and in the Indian Ocean. These correspond to the regions where we found especially frequent MHW-NPPX events in the observation-based estimates and in the two models. In addition, previous studies reported elevated chlorophyll concentrations during MHWs over regions with high nutrient concentrations, such as in the northern reaches of the southern Ocean (Hayashida et al., 2020; Gupta et al., 2020). These are regions where we also found compound events to be rare.

We then investigated the drivers of compound MHW-NPX events and the reasons why ESM2M-LE and CESM2-LE have similar compound event likelihoods in the low latitudes and divergent likelihoods in the high latitudes (third goal). We found that the models represent NPPX events of different magnitude and duration, which is suggestive of different drivers for NPPX events during MHWs. In both models, higher temperatures and circulation changes generally have a positive effect on NPP during MHW-NPPX events. In ESM2M-LE, these effects are counteracted by enhanced nutrient limitation in the low latitudes
and by enhanced light limitation in the high latitudes, which contribute to driving approximately half of the negative NPP anomaly by directly limiting phytoplankton growth. Changes in the nutrient and light limitation may also be the reason why phytoplankton loss exceeds its production over the global ocean, resulting in the buildup of a negative biomass anomaly that contributes to driving the other half of the negative NPP anomaly during MHW-NPPX events. In CESM2-LE, nutrient limitation over the global ocean is too weak to counterbalance the positive temperature effect on phytoplankton growth, though
it may moderate the growth sufficiently for NPP to be exceeded by phytoplankton loss, resulting in a biomass decrease over time. Lower biomass is the main driver of NPPX events over the global ocean in CESM2-LE. These divergent drivers of NPPX events in ESM2M-LE and CESM2-LE reflect the low degree of agreement in how ESMs represent phytoplankton growth and loss (Laufkötter et al., 2015), with this constituting a major source of uncertainties in global projections of NPP under global warming (Laufkötter et al., 2015; Frölicher et al., 2016; Kwiatkowski et al., 2020; Tagliabue et al., 2021). We expect ESMs





to differ not only in their projection of NPP, but also in how they simulate future changes in NPPX events and compound MHW-NPPX events, depending on how the drivers of NPPX events evolve under global warming.

These NPPX drivers may well also be responsible for the differences in the likelihood of compound MHW-NPPX events between the models. We expect MHWs to be frequently associated with increases in vertical stratification that inhibit the upward mixing of deep nutrients (Holbrook et al., 2019; Hayashida et al., 2020); therefore, in regions where nutrient limitation

is the dominant NPPX driver, we would expect NPPX events to frequently co-occur with MHWs. That is indeed the case in the low latitudes, where nutrient limitation drives NPPX events in the two models via its direct effect on the growth rate (in ESM2M) and its indirect effect on $NPP-Loss$, which reduces the biomass (in ESM2M-LE and CESM2-LE). Previous studies identified nutrient limitation as the main driver of negative NPP anomalies during MHWs. For example, Whitney (2015) and Le et al. (2019) found that decreased westerly winds and southward Ekman transport over the eastern part of the North Pacific

transition zone reduced nutrient concentrations during the Blob and thus inhibited NPP. Compound MHW-NPPX events are also relatively frequent in CESM2-LE in the high latitudes, where nutrient limitation contributes to driving NPPX events. On the other hand, it has been shown that MHWs are associated with enhanced incident shortwave radiation in the high latitudes (Vogt et al., 2022). Therefore, in regions where light limitation drives NPPX events, we expect rare compound events, which is indeed the case in the high latitudes in ESM2M-LE.

Our analysis revealed that compound MHW-NPPX events are accompanied with shifts in phytoplankton species. The models suggest a general shift towards larger phytoplankton over most of the global ocean during MHW-NPPX events, except in the eastern equatorial Pacific in ESM2M-LE and in CESM2-LE, as well as north of 35°N and between 35°S and 50°S in CESM2-LE, where the contribution of smaller phytoplankton to the total NPP increases during MHW-NPPX events. In general, the shift towards larger phytoplankton occurs over regions where small phytoplankton are dominant and vice-versa. Other studies

have previously documented phytoplankton shifts during MHWs (Yang et al., 2018; Wyatt et al., 2022). Wyatt et al. (2022), for example, described a relative shift towards small phytoplankton in the northeast Pacific during the 2014–2015 Blob due to a stronger response of large phytoplankton to reduced nutrient levels and a stronger response of small phytoplankton to increased light availability driven by shallower mixed layers. Small phytoplankton even increased during the Blob over the Gulf of Alaska (Wyatt et al., 2022), in agreement with CESM2-LE which simulates increased small phytoplankton NPP during

MHW-NPPX events in the northern high latitudes (Fig. 3c). Peña et al. (2019) also found a shift towards cyanobacteria, i.e., small phytoplankton, in the northeastern Pacific during the Blob. Their results are consistent with modeling studies showing that a surface ocean with lower nutrient concentrations and increased light availability favors smaller phytoplankton species (Litchman et al., 2006; Acevedo-Trejos et al., 2014). These phytoplankton shifts might lead to cascading impacts on marine ecosystems depending on which phytoplankton type marine species preferentially graze on (Cavole et al., 2016; Bindoff et al.,

2019; Cheung and Frölicher, 2020). They might also impact the biological carbon pump, as larger and heavier phytoplankton sink faster to the deep ocean (Boyd and Harrison, 1999). To better predict phytoplankton shifts and their impacts on marine ecosystems and the carbon pump during MHX-NPPX events, we need models to accurately simulate these events and their associated changes in small and large phytoplankton NPP. Yet models such as ESM2M and CESM2 still disagree, especially in the high latitudes.





One important aspect of our study is the use of large ensemble simulations (LES) with high-frequency ocean output, encompassing not only SST and NPP but also diagnostic variables used for driver attribution. The large sample size mandated by the study of compound extreme events is even larger than that required for extreme events with single variables (Deser et al., 2020; Burger et al., 2021; Zscheischler and Lehner, 2022). This is particularly important under non-stationary conditions, when relatively short time series need to be analysed to obtain a picture of quasi-stationary conditions. The application of two

different Earth system models facilitated an exploration of how uncertainties in the formulation of NPP manifest themselves in the occurrence (pattern and frequency) of compound events. This should complement work by Kwiatkowski et al. (2020) and Bopp et al. (2021) in underscoring the challenges faced by the Earth system modeling community given pervasive NPP uncertainty.

One challenging aspect of our study is the lack of agreement between observation-based estimates of the frequency of com-
pound MHW-NPPX events in the mid to high southern latitudes, which makes it difficult to determine whether the ESMs well represent compound MHW-NPPX events and their drivers over this region. NPP estimates produced by models assimilating satellite data are still uncertain and highly sensitive to their respective model configurations (e.g. Behrenfeld et al. (2005); Long et al. (2021a)), especially in sea-ice covered regions. We decided to include five observation-based NPP products in this study to take into account the high uncertainty in NPP estimates, which affects the observation-based estimates of MHW-NPPX event
frequency. Direct NPP measurements would be needed to better constrain the NPP estimated by ESMs in the future.

To conclude, the combination of a MHW and an NPPX event constitutes a compound hazard which potentially leads to severe impacts on marine organisms and ecosystems. Here, we assessed whether LES from two ESMs can be used to understand compound MHW-NPPX events in the ocean and to project them into the future. Our analysis reveals that the likelihood of compound MHW-NPPX events depends on how ESMs represent the factors limiting phytoplankton growth and loss. These
factors are similar in ESM2M and CESM2 in the low latitudes but differ in the high latitudes. This identifies an important need for improved process understanding in the models used for predicting and projecting the potentially harmful compound events in the ocean.

## Appendix A: Ecosystem model description

### A1    GFDL ESM2M: TOPAZv2

TOPAZv2 stands for Tracers of Ocean Phytoplankton with Allometric Zooplankton version 2.0. It is the biogeochemical and ecological module used in GFDL's ESM2M (Dunne et al., 2013). Three phytoplankton types are represented: nano- (or small) phytoplankton, large phytoplankton and diazotrophs. Nitrogen in each phytoplankton type $i$ is a prognostic variable.

$$\partial_t n_i = -\nabla(\overrightarrow{u}.n_i) + NPP_i - Loss_i \tag{A1}$$

where $\nabla(\overrightarrow{u}.n)$ corresponds to the physical advection and mixing of phytoplankton nitrogen $n$ by ocean circulation, $NPP$ is
the nitrogen-specific NPP, and $Loss$ is the nitrogen-specific decay. NPP of each phytoplankton type is the product of its growth





rate $\mu$ and its biomass $n$:

$$NPP_i = \mu_i n_i \tag{A2}$$

### A1.1 Phytoplankton growth

In TOPAZv2, the nitrogen-specific growth rate is defined for all phytoplankton types as follows:

$$\mu_i = \frac{\mu_{max} N_{lim_i} T_f + \varepsilon}{1 + \zeta} L_{lim_i} \approx \mu_{max} N_{lim_i} L_{lim_i} T_f \tag{A3}$$

where $N_{lim}$ is the nutrient limitation, $L_{lim}$ is the light limitation, and $T_f$ is an Eppley function of the temperature.

### A1.2 Nutrient limitation

$N_{lim}$ is computed using Leibig's law of the minimum, where $N_{Fe}$, $N_{Si(OH)_4}$, $N_{PO_4}$, $N_{NH_4}$, and $N_{NO_3}$ correspond to the nutrient limitation specific to iron, silicon, phosphate, ammonia, and nitrate.

$$N_{lim_i} = \min(N_{Fe_i}, N_{Si(OH)_{4_i}}, N_{PO_{4_i}}, N_{NH_{4_i}} + N_{NO_{3_i}})$$

Nutrient limitation is represented according to Michaelis-Menten kinetics, where $K_{Fe}$, $K_{Si(OH)_4}$, $K_{PO_4}$, $K_{NH_4}$, and $K_{NO_3}$ are the half-saturation constants of each nutrient.

$$N_{Fe_i} = \frac{Fe}{Fe + K_{Fe_i}} \tag{A4}$$

$$N_{PO_{4_i}} = \frac{PO_4}{PO_4 + K_{PO_{4_i}}} \tag{A5}$$

$$N_{Si(OH)_{4_i}} = \frac{Si(OH)_4}{Si(OH)_4 + K_{Si(OH)_{4_i}}} \tag{A6}$$

$$N_{NH_{4_i}} = \frac{NH_4}{NH_4 + K_{NH_{4_i}}} \tag{A7}$$

Nitrate limitation with ammonia inhibition is represented after Frost and Franzen (1992).

$$N_{NO_{3_i}} = \frac{NO_3}{NO_3 + K_{NO_{3_i}}} \cdot (1 + \frac{NH_4}{K_{NH_{4_i}}}) \tag{A8}$$





### A1.3  Light limitation

Light limitation is calculated as:

$$L_{lim_i} = 1 - e^{-\dfrac{\alpha_i \theta_i Irr}{N_{lim_i} T_f \mu_{max} + \varepsilon}} \tag{A9}$$

where $\alpha$ is the light harvest coefficient, $\theta$ is the chlorophyll to carbon ratio and $Irr$ corresponds to mean light level (W m$^{-2}$) of a depth layer. $\mu_{max}$ is the maximal growth rate and $\varepsilon$ a constant for numerical stability. More details on how to compute $\theta$, $N_{Fe}$, $N_{PO_4}$, and the limitation terms specific to iron and phosphate when Fe:N or P:N vary in phytoplankton are given in Dunne et al. (2013).

### A1.4  Temperature function

The temperature function is given as:

$$T_f = e^{K_{epp} T} \tag{A10}$$

where T is the temperature and $K_{epp}$ is the constant temperature coefficient for growth.

### A1.5  Phytoplankton grazing

In TOPAZ, phytoplankton decays through grazing only. Grazing is computed separately for small and large phytoplankton.

$$G_{small} = min(k_{graz_{max}}, \lambda_0 T_f \frac{n_{small}}{n\star}) \frac{n_{small}^2}{n_{small} + n_{min}} \tag{A11}$$

$$G_{large} = min(k_{graz_{max}}, \lambda_0 T_f (\frac{n^{graz_{large}}}{n\star})^{1/3} \frac{n^{graz_{large}}}{n_{large} + n_{min}}) n_{large} \tag{A12}$$

where $k_{graz_{max}}$ is the maximum grazing rate, $\lambda_0$ is another grazing rate, and $n\star$ is the pivot phytoplankton concentration for grazing-based variations in ecosystem structure. $n^{graz_{large}}$ is an implicit phytoplankton concentration after incorporation of a temperature-dependent time lag:

$$n^{graz_{large}} = (n^{graz_{large}})_{old} . e^{\frac{n_{large} - (n^{graz_{large}})_{old}}{n_{large} + (n^{graz_{large}})_{old}} . 2 . min(1, T_f \frac{\Delta t}{\tau})} \tag{A13}$$

where $(n^{graz_{large}})_{old}$ corresponds to $n^{graz_{large}}$ of the previous time step $\Delta t$, and $\tau$ is the temperature-dependent response timescale for grazers, which is set to a very small number to simulate instantaneous response. More explanations are given in Dunne et al. (2013).





| Parameter | Value | Unit | Name |
|-----------|-------|------|------|
| $K_{epp}$ | 0.063 | $°C^{-1}$ | temperature coefficient for growth |
| $\mu_{max}$ | 1.5e-5 | $s^{-1}$ | maximal growth rate at 0°C |
| $\zeta$ | 0.1 | - | photosynthetic respiration loss |
| $\varepsilon$ | 1e-30 | $s^{-1}$ | constant for numerical stability |
| $\alpha$ | 9.2e-5 | g C (g Chl)$^{-1}$ m$^2$ W$^{-1}$ s$^{-1}$ | light harvest coefficient |
| $n\star$ | 1.9e-6 * 16.0 / 106.0 | mol N kg$^{-1}$ | pivot phytoplankton concentration |
| $k_{graz_{max}}$ | 6 | day$^{-1}$ | maximum phytoplankton grazing rate |
| $\lambda_0$ | 0.19 | day$^{-1}$ | phytoplankton grazing rate |
| $n_{min}$ | 1e-6 | mol N m$^{-3}$ | minimum phytoplankton concentration for grazing |
| $\tau$ | 0.01 | day$^{-1}$ | temperature-dependent response timescale for grazers |

**Table A1.** Parameter values used in TOPAZv2 to compute the growth rate of both small and large phytoplankton

| Parameter | Value | Unit | Name |
|-----------|-------|------|------|
| $K_{Fe}$ | 3e-9 | mol dissolved Fe kg$^{-1}$ | half saturation coefficient |
| $K_{PO_4}$ | 2e-7 | mol PO$_4$ kg$^{-1}$ | half saturation coefficient |
| $K_{NH_4}$ | 2e-7 | mol NH$_4$ kg$^{-1}$ | half saturation coefficient |
| $K_{NO_3}$ | 2e-6 | mol NO$_3$ kg$^{-1}$ | half saturation coefficient |

**Table A2.** Parameter values used in TOPAZv2 to compute the growth rate of small phytoplankton

| Parameter | Value | Unit | Name |
|-----------|-------|------|------|
| $K_{Fe}$ | 9e-9 | mol dissolved Fe kg$^{-1}$ | half saturation coefficient |
| $K_{PO_4}$ | 6e-7 | mol PO$_4$ kg$^{-1}$ | half saturation coefficient |
| $K_{NH_4}$ | 6e-7 | mol NH$_4$ kg$^{-1}$ | half saturation coefficient |
| $K_{NO_3}$ | 6e-6 | mol NO$_3$ kg$^{-1}$ | half saturation coefficient |
| $K_{Si(OH)_4}$ | 1e-6 | mol Si(OH)4 kg$^{-1}$ | half saturation coefficient |

**Table A3.** Parameter values used in TOPAZv2 to compute the growth rate of large phytoplankton






## A2 CESM2: MARBL

The Marine Biogeochemistry Library (MARBL) is the biogeochemical component of CESM2. It is a prognostic ocean biogeochemistry model that simulates marine ecosystem dynamics and the coupled cycles of carbon, nitrogen, phosphorus, iron, silicon, and oxygen (Long et al., 2021b). Three phytoplankton types are represented: small phytoplankton, diatoms and dia-

zotrophs. The concentration $P_i$ of each phytoplankton type $i$ is a prognostic variable.

$$\partial_t P_i = -\nabla(\overrightarrow{u}.P_i) + NPP_i - Loss_i \tag{A14}$$

where $\nabla(\overrightarrow{u}.P)$ corresponds to the physical advection and mixing of phytoplankton by ocean circulation, and $Loss$ correspoonds to phytoplankton decay. NPP of each phytoplankton type is the product of its growth rate $\mu$ and its biomass $n$:

$$NPP_i = \mu_i P_i \tag{A15}$$

### 625 A2.1 Phytoplankton growth

In MARBL, the carbon-specific growth rate of phytoplankton is defined as:

$$\mu_i = \mu_{ref} N_{lim_i} L_{lim_i} T_f \tag{A16}$$

where $\mu_{ref}$ is a constant accounting for the maximum growth rate at the reference temperature of 30°C. $N_{lim}$ is the nutrient limitation, $L_{lim}$ is the light limitation, and $T_f$ is the temperature function:

### 630 A2.2 Nutrient limitation

$N_{lim}$ is computed using Leibig's law of the minimum, where $N_{Fe}$, $N_{SiO_3}$, $N_{PO_4}$, $N_{NH_4}$, and $N_{NO_3}$ correspond to the nutrient limitation specific to iron, silicon, phosphate, ammonia, and nitrate.

$$N_{lim_i} = min(N_{Fe_i}, N_{SiO_{3_i}}, N_{P_i}, N_{NH_{4_i}} + N_{NO_{3_i}}) \tag{A17}$$

$$N_{Fe_i} = \frac{Fe}{Fe + K_{Fe_i}} \tag{A18}$$

$$N_{SiO_{3_i}} = \frac{SiO_3}{SiO_3 + K_{SiO_{3_i}}} \tag{A19}$$

Phytoplankton can alternatively assimilate nitrate and ammonium following **?**, such that:

$$N_{NH_{4_i}} = \frac{\dfrac{NH_4}{K_{NH_{4_i}}}}{1 + \dfrac{NO_3}{K_{NO_{3_i}}} + \dfrac{NH_4}{K_{NH_{4_i}}}} \tag{A20}$$





$$N_{NO_{3_i}} = \frac{\dfrac{NO_3}{K_{NO_{3_i}}}}{1 + \dfrac{NO_3}{K_{NO_{3_i}}} + \dfrac{NH_4}{K_{NH_{4_i}}}}$$ (A21)

Phytoplankton is able to assimilate phosphorus in the form of phosphate (PO$_4$) and semi-labile dissolved organic phosphate (DOP); a similar approach is used to compute $N_P$.

### A2.3 Light limitation

The light limitation is given as:

$$L_{lim_i} = 1 - e^{-\dfrac{\alpha_i \theta_i Irr}{N_{lim_i} T_f \mu_{ref}}}$$ (A22)

where $\alpha$ is the light harvest coefficient, $\theta$ is the chlorophyll to carbon ratio and $Irr$ corresponds to the photosynthetically available radiation, defined as 45% of incoming short wave radiation (W m$^{-2}$). In the high latitudes, CESM2 simulates a subgrid-scale sea-ice thickness distribution and computes shortwave penetration independently in each sub-column. MARBL

then takes an area-weighted average across sub-columns to compute the grid cell mean light level. For more details on how to compute $\theta$ and $N_P$, see Long et al. (2021b).

### A2.4 Temperature function

The temperature function is given as:

$$T_f = 1.7^{\dfrac{T - 30°C}{10°C}}$$ (A23)

where T is the temperature.

### A2.5 Phytoplankton loss

In MARBL, phytoplankton decays through grazing $G$, mortality $M$ and aggregation $A$, which refers to the process by which dying phytoplankton form aggregates that sink through the water column. The three loss terms depend on $P'$, the phytoplankton concentration in excess of a temperature and depth-dependent threshold (Long et al., 2021b).

$$P'_i = max(P_i - P_{threshold_i}, 0)$$ (A24)

$$Loss_i = G(P'_i)_i + M(P'_i) + A_{i_i}(P'_i)$$ (A25)





Grazing by zooplankton is given as:

$$G_i(P'_i) = g_{max_i} T_f \frac{P'_i}{K^P + P'_i} z \tag{A26}$$

where $g_{max}$ is the maximum grazing rate, $K^P$ is the half saturation constant for phytoplankton grazing and $z$ is the zooplankton biomass.

Mortality is given as:

$$M(P'_i) = m T_f P'_i \tag{A27}$$

where m is the linear mortality rate.

Finally, aggregation is parameterized as:

$$A_i(P'_i) = a_i * (P'_i)^{1.75} \tag{A28}$$

where a is the aggregation rate (see (Long et al., 2021b) for more details).

| Parameter | Value | Unit | Name |
|---|---|---|---|
| $\mu_{ref}$ | 5 | day$^{-1}$ | resource-unlimited growth rate |
| $\alpha_{small}$ | 0.39 | mol C (g Chl)$^1$ m$^2$ W$^{-1}$ day$^{-1}$ | light harvest coefficient |
| $\alpha_{large}$ | 0.28 | mol C (g Chl)$^{-1}$ m$^2$ W$^{-1}$ day$^{-1}$ | light harvest coefficient |
| $K^P$ | 1.2 | mmol m$^{-3}$ | half saturation coefficient for grazing |
| m | 0.1 | day$^{-1}$ | Linear mortality rate |

**Table A4.** Parameter values used in MARBL to compute the growth rate of small and large phytoplankton

| Parameter | Value | Unit | Name |
|---|---|---|---|
| $K_{Fe}$ | 3e-5 | mmol dissolved Fe kg$^{-1}$ | half saturation coefficient |
| $K_{PO_4}$ | 0.01 | mmol PO$_4$ m$^{-3}$ | half saturation coefficient |
| $K_{NH_4}$ | 0.01 | mmol NH$_4$ m$^{-3}$ | half saturation coefficient |
| $K_{NO_3}$ | 0.25 | mmol NO$_3$ m$^{-3}$ | half saturation coefficient |
| g$_{max}$ | 3.3 | day$^{-1}$ | Maximum grazing rate |

**Table A5.** Parameter values used in MARBL to compute the growth rate of small phytoplankton





| Parameter | Value | Unit | Name |
|:---:|:---:|:---:|:---:|
| $K_{Fe}$ | 7e-5 | mmol dissolved Fe kg$^{-1}$ | half saturation coefficient |
| $K_{PO_4}$ | 0.05 | mmol PO$_4$ m$^{-3}$ | half saturation coefficient |
| $K_{NH_4}$ | 0.05 | mmol NH$_4$ m$^{-3}$ | half saturation coefficient |
| $K_{NO_3}$ | 0.5 | mmol NO$_3$ m$^{-3}$ | half saturation coefficient |
| $K_{SiO_3}$ | 0.7 | mmol SiO3 m$^{-3}$ | half saturation coefficient |
| g$_{max}$ | 3.15 | day$^{-1}$ | Maximum grazing rate |

**Table A6.** Parameter values used in MARBL to compute the growth rate of large phytoplankton






## Appendix B: Additional figures



**Figure B1.** Climatological mean and standard deviation of the observation-based NPP estimates (mg C m$^{-2}$ day$^{-1}$) calculated from NOBM (a,b), Standard-VGPM (c,d), Eppley-VGPM (e,f), CbPM (g,h) and CAFE (i,j), and simulated by ESM2M-LE (k,l) and CESM-LE (m,n) over 1998-2018. Grey boxes indicate the globally integrated mean NPP (Pg C year$^{-1}$). We use 5-day mean NPP output for all products except for the VGPM-based products (c-j).





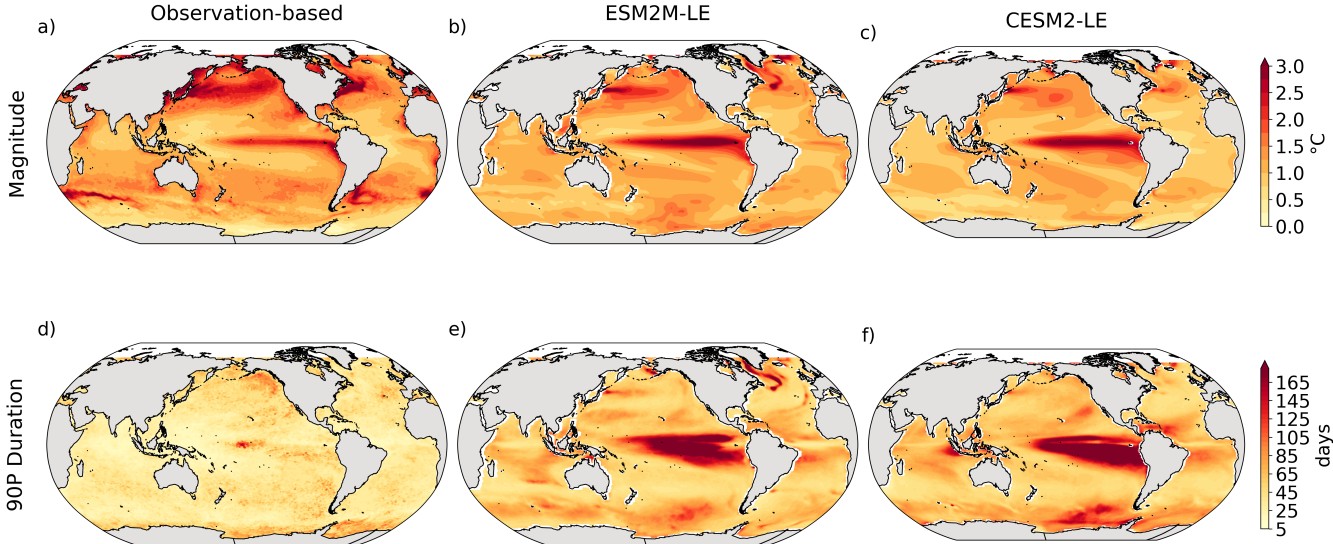

**Figure B2.** Simulated mean magnitude and duration of MHWs over 1998-2018. Mean SST anomaly relative to the seasonal cycle ($^\circ$C) during MHWs in (a) the observation-based estimate, (b) ESM2M-LE and (c) CESM2-LE. Simulated 90th percentile of the MHW durations (days) in the (d) observation-based estimate, (e) ESM2M-LE and (f) CESM2-LE. The global mean magnitude equals 1.3, 1.2 and $1.2^\circ C$, while the global mean 90th percentile of the duration equals 36, 69 and 75 days in the observations, ESM2M-LE and CESM2-LE, respectively.





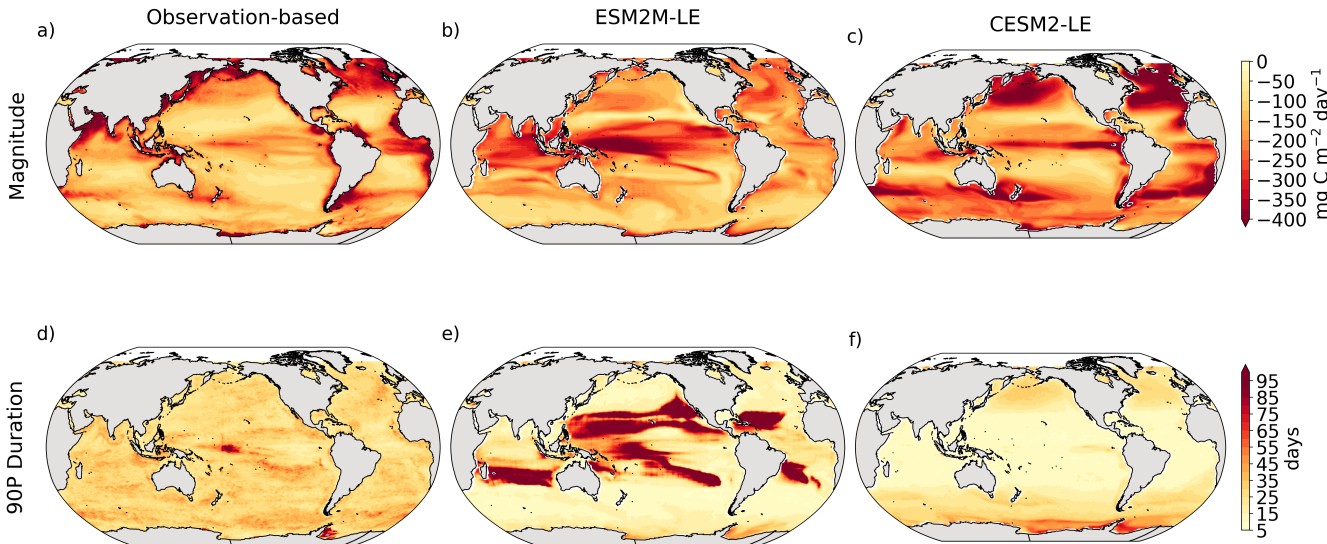

**Figure B3.** Simulated mean magnitude and duration of NPPX events over 1998-2018. Mean NPP anomaly relative to the seasonal cycle (mg C m$^{-2}$ day$^{-1}$) during NPPX events in (a) observation-based estimates, (b) ESM2M-LE and (c) CESM2-LE. 90th percentile of the NPPX events duration (day) in (d) observation-based estimates, (e) ESM2M-LE and (f) CESM2-LE. The global mean magnitude equals -209, -182 and -223 mg C m$^{-2}$ day$^{-1}$, while the global mean 90th percentile of the duration equals 29, 34 and 18 days in the observations, ESM2M-LE and CESM2-LE, respectively.

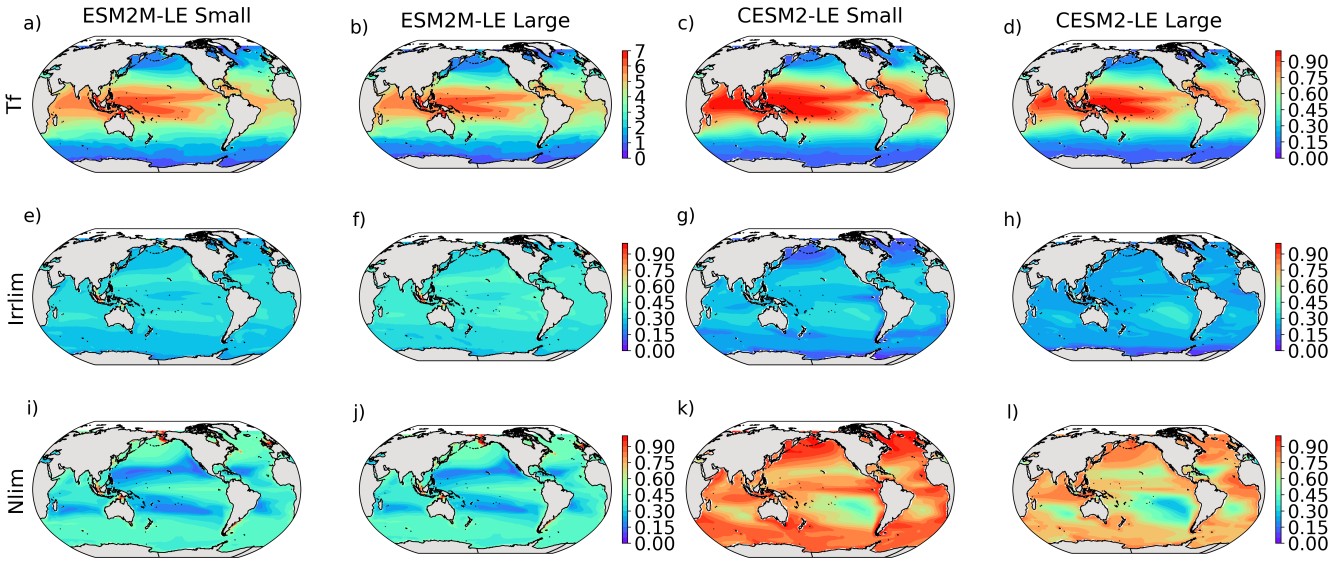

**Figure B4.** Simulated mean states of the temperature (a-d, $T_f$), light (e-h, $Irrlim$) and nutrient (i-l, $Nlim$) limitations on the small and large phytoplankton growth rates in ESM2M-LE and CESM2-LE over 1998-2018.





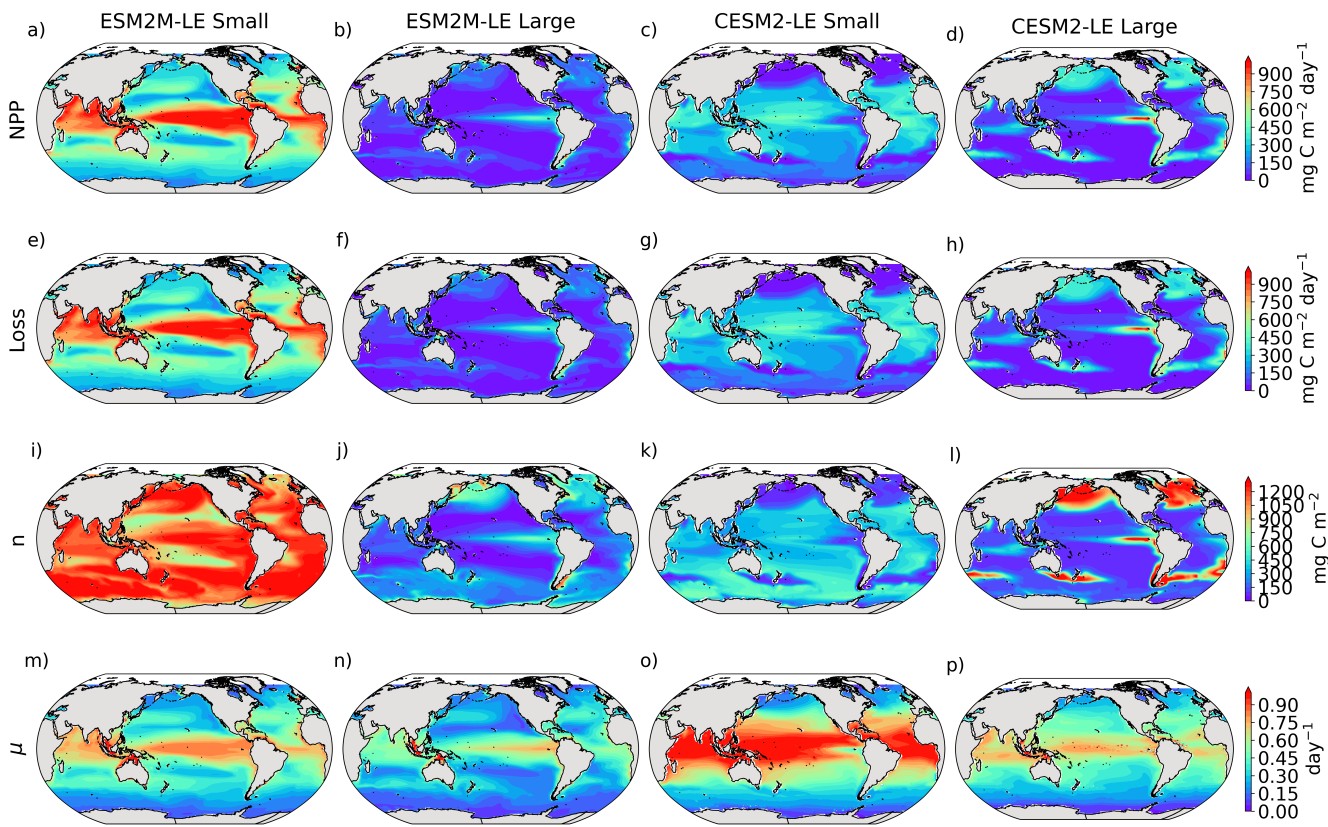

**Figure B5.** Simulated mean states of small and large phytoplankton NPP (a-d, mg C m$^{-2}$ day$^{-1}$), loss (e-h, mg C m$^{-2}$ day$^{-1}$), biomass (i-l, mg C m$^{-2}$) and growth rate (m-p, day$^{-1}$) in ESM2M-LE and CESM2-LE over 1998-2018.



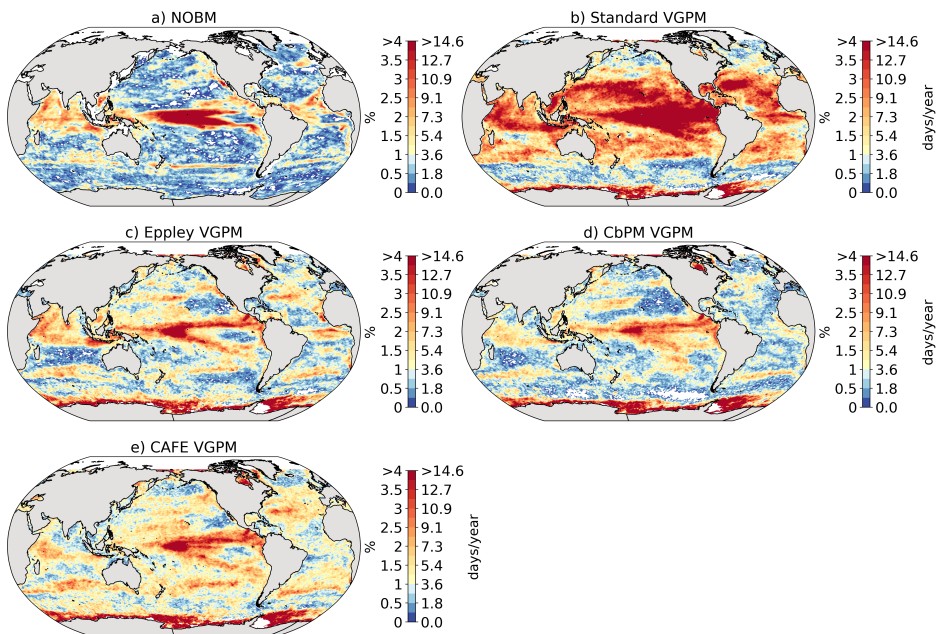

**Figure B6.** Likelihood (%) of compound MHW-NPPX events estimated using the observation-based NPP product of (a) NOBM, (b) Standard-VGPM, (c) Eppley-VGPM, (d) CbPM-VGPM, and (e) CAFE-VGPM.

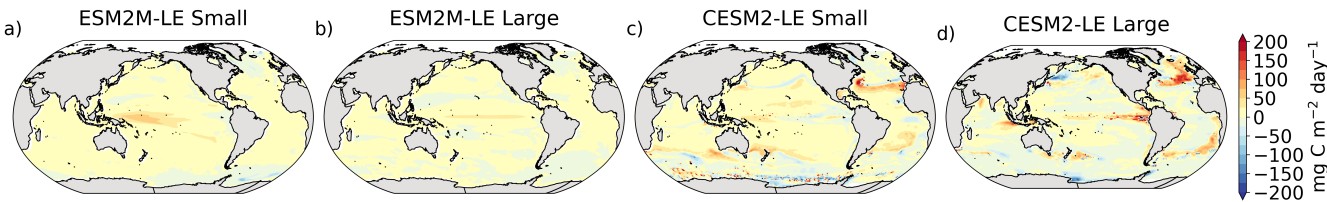

**Figure B7.** Difference between the NPP anomaly $dNPP$ during compound MHW-NPPX events and its decomposition into a contribution of the growth rate anomaly $nd\mu$ and of the biomass anomaly $\mu dn$ (mg C m$^{-2}$ day$^{-1}$) during compound MHW-NPPX events for small and large phytoplankton in ESM2M-LE (a,b) and in CESM2-LE (c,d).





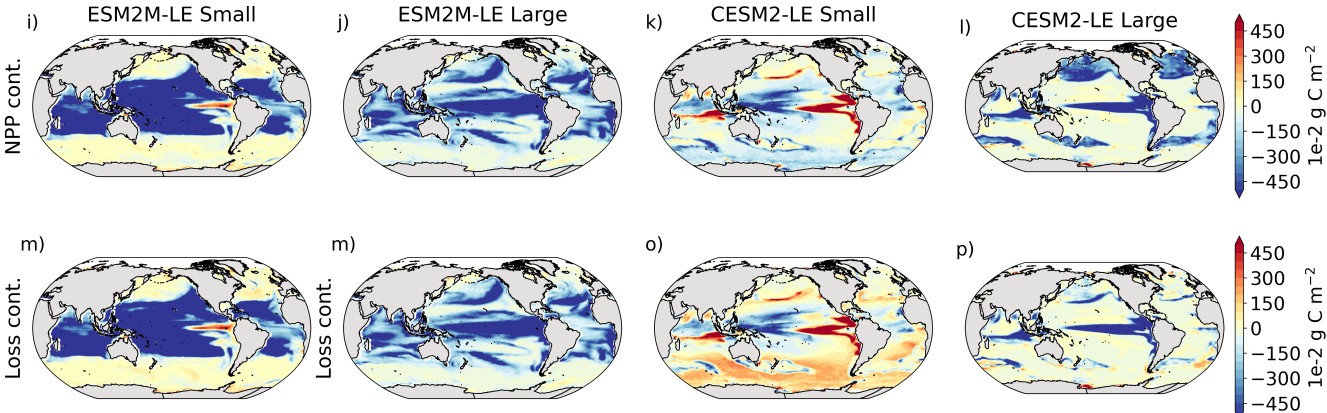

**Figure B8.** Contributions (mg C m$^{-2}$ day$^{-1}$) of small and large phytoplankton $NPP$ and $Loss$ to the buildup of the maximum biomass anomaly (mg C m$^{-2}$ day$^{-1}$) in ESM2M-LE (a,b,e,f) and in CESM2-LE (c,d,g,h).

*Author contributions.* NLG, JZ and TLF designed the study. KR and RT provided the CESM2 output. NLG performed the analysis and wrote the initial draft of the manuscript. All authors discussed the analysis and results, and contributed to the writing of the paper.

*Data availability.*

*Competing interests.* All authors declare no competing interests.

*Disclaimer.* The work reflects only the authors' view; the European Commission and their executive agency are not responsible for any use that may be made of the information the work contains.

*Acknowledgements.* NLG is funded by the Alfred Bretscher Fund. TLF acknowledges support from the Swiss National Science Foundation (PP00P2-198897) and from the European Union's Horizon 2020 research an innovation programme under grant agreement no. 820989

(project COMFORT) and no. 862923 (project AtlantECO). KBR was supported by the Institute for Basic Sciences (IBS), Republic of Korea, under IBS-R028-D1. The GFDL ESM2M simulations were conducted on the Swiss National Supercomputing Centre and the CESM2-LE was run on the IBS/ICCP supercomputer 'Aleph' in South Korea. The authors thank Friedrich Burger for the help with setting up the simulations, Cecile Rousseaux for providing the NASA Ocean Biogeochemical Model dataset for net primary production, and Charlotte Laufkötter for initial discussions.



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
