# Peer review of "Hotspots and drivers of compound marine heatwave and low net primary production extremes"

_EGUsphere, 2022_

## Referee Comment (RC1)

This is an interesting, well-written article that explores the incidence and drivers of compound marine extreme events (high temperature and low primary production conditions) in observations and two ESMs. The authors focus on the drivers of low NPP extremes during marine heat waves and use an interesting approach to decompose the relative contribution of these drivers in each model. Their work further highlights the progress that needs to be made to robustly "observe" and model marine primary production.

My main comments are that I think the authors could:

1. Better explain the decomposition of phytoplankton biomass anomalies. Specifically, the advection/mixing terms which appear to be very important to the decomposition of NPP anomalies in Figure 6.

2. Explore and further discuss the mechanisms behind some of the non-intuitive NPP responses. I find the increase in light limitation in the high latitudes particularly non-intuitive in ESM2M-LM.

3. Asses how the drivers of low NPP extremes differ in a MHW as opposed to non-MHW conditions. I.e. Is there anything different/unique about the drivers of low NPP extremes when they occur within a MHW?

In addition to this I have a number of minor comments below that will hopefully improve manuscript clarity and accessibility. Subject to these comments being addressed, I am happy to recommend publication.

L40 Is it fair to characterize 'the blob' as solely a MHW/NPPX event what about cooccurring deoxygenation and acidification?

L74 A word seems missing after "phytoplankton". Growth, biomass, both?

L157 interval(s)

L161 Maybe clarify you mean "ocean ecology/biogeochemistry. The models presumably also have differing land ecosystem models/DGVMs.

L166 Are you sure this is correct? In the appendices it appears that MARBL uses higher half saturation constants. Maybe there's an error in the appendix legends or units. To aid comparison I would recommend using the same units for parameters that are present in each model where possible.

L167 Are nutrient levels similar between models? I ask because this is an interpretation that the authors come back to.

L182 Perhaps add something here on the rationale behind deseasonalizing. Do observations support this approach? It seems contrary to the thermal stress DHW estimates that are used for coral reefs.

L186. Over what period are percentiles computed? Does this correspond to a 1-in-10-day event? This seems like a low threshold for defining an extreme that probably need to be discussed/justified.

L193. I'm not sure I would intrepet these as hotspots of compound extremes, rather simply a lack of independence between the drivers of MHW and NPPX events. Warming/stratification/reduced nutrient supply?

L196 I find it a little strange to have model evaluation in the methods but I suppose there's a certain logic to not having this in the results.

Figure 1. Maybe mention that Std is normalized on the Taylor plot axes.

Figure 1d. One wonders if an overestimation of durations is a resolution issue. E.g. a result of non-eddying models failing to capture short-lived extremes that maybe associated with mesoscale processes.

L232 This may be an overstatement for NPP where the models seem a long way from the ref (even if there is very little confidence in the ref).

L266 Is this advection/mixing term explicitly calculated or is this a residual? If the former can the authors say how well the decomposition works? Related to this, are the driver decompositions calculated online or using models outputs at a certain resolution?

L269 The method could be clarified here. Do you mean the maximum absolute anomaly? And if so, relative to what, the climatological mean?

Results- Lots of the interpretation and speculation that is currently in the results section is really "discussion". I would recommend having a combined results and discussion section to avoid substantial reformatting.

Figure 3. It's not entirely clear to me how to interpret the contours in this figure. Should there be values associated with the contours?

L332. You're showing NPP not "abundance" (although they are presumably closely related in the climatological mean).

L345-347 I suggest removing the "contribution" phrasing here which is confusing.

L356 Any such NPP increases are very difficult to see with the current color scale of Figure 3. Maybe you want to white-center your color scale to avoid overinterpretation of very low anomalies.

L364 Is the "per day" unit correct? I.e is this a rate of decline in nutrient limitation? I would expect a mean nutrient limitation anomaly to be unitless.

L373-374 I find it odd that the impact of nutrient limitation isn't exacerbated in the larger phytoplankton given their higher half saturation constants. Could the authors comment on this?

L366 As mentioned previously, why are MHW-NPPX events associated with enhanced light limitation in the high latitudes? Presumably these waters are more stratified or are sea-ice dynamics interfering with the expectation that mixing of phytoplankton blow the euphotic zone declines during a MHW?

Figure 4. Perhaps the authors could add a comment in the legend on how well the growth rate decomposition works, assuming it is not perfect.

L381 See earlier comment on this.

Figure 5. The units in the figure and legend don't match.

L421. Maybe clarify you mean the grazing of phytoplankton by zooplankton.

Figure 6. I really like the idea behind this figure but it would be nice to see a sum of the contributions to see how well the decomposition works in each region. What does the "+res." term stand for (residual?). As previously stated, I am uncertain how to interpret these large circulation anomalies that offset NPP declines in all regions.

Figure 6. I'm quite surprised by the very small Nlim term in CESM2, particularly in the low latitudes and the eastern equatorial Pacific. Can the authors comment on this.

L443-444 Don't the models basically show no light limitation effect in the low latitudes? Perhaps best to refer to light limitation and not light levels. Presuming it's the vertical distribution of phytoplankton biomass and not incoming shortwave which is the main driver of light limitation during MHWs (unless authors have assessed this?).

L450 See previous comment.

L457 Is there causality here? Do higher temperatures enhance loss or are they simply associated with enhanced loss?

L478 I don't think this citing of the study "goals" is needed and is a bit distracting.

L496-497 Presumably nutrient and light limitation only influence production not loss. Can the authors discuss how loss is actually affected? Is there greater thermal sensitivity of grazers than phytoplankton? how do changes in physical dynamics influence loss during MHWs?

---

## Author Comment (AC3)

**Hotspots and drivers of compound marine heatwave and low net primary production extremes**
**Response to reviewers comments**

Natacha Le Grix, Jakob Zscheischler, Keith B. Rodgers,
Ryohei Yamaguchi, and Thomas L. Frölicher

October 24, 2022
* * *
**1 Response to Lester Kwiatkowski**

**1.1 General Comments**

This is an interesting, well-written article that explores the incidence and drivers of compound marine extreme events (high temperature and low primary production conditions) in observations and two ESMs. The authors focus on the drivers of low NPP extremes during marine heat waves and use an interesting approach to decompose the relative contribution of these drivers in each model. Their work further highlights the progress that needs to be made to robustly "observe" and model marine primary production.

We thank the reviewer for the positive feedback and his detailed comments on our manuscript.

**1.2 Main Comments**

My main comments are that I think the authors could:

1. Better explain the decomposition of phytoplankton biomass anomalies. Specifically, the advection/mixing terms which appear to be very important to the decomposition of NPP anomalies in Figure 6.

In the Methods section 2.5, we reformulated the description of the drivers of biomass changes:

"Drivers of a change in phytoplankton biomass $n$ are less trivial, as $n$ depends on NPP itself. In TOPAZv2 and MARBL, $n$ is considered a tracer whose time derivative is defined as follows:
$$\partial_t n_i = NPP_i - Loss_i + Circ_i \tag{1}$$

where $NPP$ and $Loss$ are the biological production and decay of phytoplankton, respectively, and $Circ$ corresponds to the physical advection and mixing of phytoplankton by ocean circulation. The models equations only hold at the time and vertical resolution of model computations, i.e, at 2-hour and 10-meter resolution. Here we use 5-day mean output and data averaged over the top 100-meter layer. Therefore, equation 1 becomes:

$$\partial_t n_i = NPP_i - Loss_i + Circ_i + Errors_i \tag{2}$$

Given that we do not have the necessary output to compute the circulation term, we cannot assess how small the $Errors$ are, and therefore we cannot neglect them.

Over time, biomass changes build up a biomass anomaly $dn$ that might be sufficient to drive or contribute to driving extremely low $dNPP$. In this study, we intend to explain the contribution of $dn$ to $dNPP$ during compound MHW-NPPX events using equation 2. We integrate $\partial_t n$, $NPP$ and $Loss$ over all periods over which $dn$ builds up, i.e., over which $n$ changes from its climatological mean value (at $t_1$) to its maximum absolute anomaly relative to the climatology reached during a compound event (at $t_2$). $\Delta n$ refers to the integrated biomass change.

$$dn_i \sim \Delta n_i = \int_{t_1}^{t_2} \partial_t n_i dt = \int_{t_1}^{t_2} (NPP_i - Loss_i) dt + \int_{t_1}^{t_2} (Circ_i + Errors_i) dt \tag{3}$$

$$dn_i \sim \Delta n_i = \int_{t_1}^{t_2} \partial_t n_i dt = \int_{t_1}^{t_2} (NPP_i - Loss_i) dt + Residual_i \tag{4}$$

The term $\int_{t_1}^{t_2} (NPP_i - Loss_i) dt$ accounts for the contribution of biological processes to $\Delta n$, whereas $Residual$ includes both the contribution of ocean circulation to $\Delta n$ and all errors inherent to the decomposition using 5-day mean and vertically integrated output. Results are averaged over all compound events. In theory, this method would enable us to quantify the contribution of biological processes to $dn$. However, errors in the decomposition might be substantial and result in a poor estimation of that contribution. Further work with higher temporally and spatially resolved model output is needed to fully decompose the biomass changes during compound MHW-NPPX events into its drivers.

Details on the computation of phytoplankton Loss are provided in Sections A1.5 for TOPAZv2 and A2.5 for MARBL, and Fig. B5 presents the climatological mean states of $NPP$, $Loss$, $n$ and $\mu$." (l. 274-299 in the revised manuscript)

In the first version of this paper, we had assumed the $Errors$ to be negligible and used equation 4 to explain biomass changes. We revised over-interpretations of our results in the whole manuscript. The circulation term is now considered as a residual term, and the decomposition in equation 4 simply used as a tool to better understand the sign of biomass changes, rather than a way to quantify the exact contributions of biology and circulation to changes in biomass.

2. Explore and further discuss the mechanisms behind some of the non-intuitive NPP responses. I find the increase in light limitation in the high latitudes particularly nonintuitive in ESM2M-2M.

We added the following paragraph to section 3.3.1:

"Even though the light limitation depends on a number of factors other than the light supply, such as temperature, nutrient levels, mixed layer depth or the carbon to chlorophyll ratio in phytoplankton, increased light limitation is here a direct result of reduced light supply by -13 $Wm^{-2}$ on average (Fig. 6a). High latitude MHWs are, however, mainly driven by enhanced shortwave radiation in summer (see Fig. 4 in Vogt et al. [2022]). Enhanced shortwave radiation seem incompatible with reduced light levels, hence the low compound MHW-NPPX event frequency in the high latitudes in ESM2M-LE. Therefore, for MHWs to co-occur with reduced light levels, they must be driven by other drivers than radiative heating, such as vertical mixing or advective processes. These drivers might be compatible with clouds or extended sea-ice cover, and thus with light limitation. In addition, high temperatures during MHWs also raise energy demand on phytoplankton and directly enhance the light limitation (see the role of $T_f$ in sections A.1.3 and A.2.3)." (l. 394-402)

3. Assess how the drivers of low NPP extremes differ in a MHW as opposed to non-MHW conditions. I.e. Is there anything different/unique about the drivers of low NPP extremes when they occur within a MHW?

We did some additional analyses to address this question. We found that similar growth rate and biomass anomalies drive low NPP during NPPX events as during MHW-NPPX events (Figure 1 below compared to Figure 3 of the manuscript).

[Figure]

Figure 1: Small and large phytoplankton NPP anomalies $dNPP$ relative to the climatological seasonal cycle (mg C m$^{-2}$ day$^{-1}$) during NPPX events in ESM2M-LE (a,b) and in CESM2-LE (c,d), and contributions of the growth rate $nd\mu$ (e-h) and of the biomass anomaly $\mu dn$ (i-l) to these NPP anomalies. Contours on panels a-d indicate the climatological mean state of small and large NPP averaged over 1998-2018 (see also Fig. B5a-d); labels have been omitted.

However, we found different drivers for the growth rate anomaly in each case. The nutrient limitation on phytoplankton growth in the low to mid latitudes is especially strong during MHW-NPPX events compared to simple NPPX events (Figure 2 below compared to Figure 4 of the manuscript). Stronger nutrient limitation counteracts the positive temperature effect on phytoplankton growth during MHWs in these regions.

[Figure]

Figure 2: Growth rate anomaly (day$^{-1}$) of small and large phytoplankton during NPPX events in ESM2M-LE (a,b) and in CESM2-LE (c,d), and contributions of a change in the temperature function (e-h), in the light limitation (i-l) and in the nutrient limitation (m-p) to this growth rate anomaly.

We added the following sentence to section 3.3.1: "In both models, nutrient limitation on phytoplankton growth is especially strong during MHW-NPPX events compared to simple NPPX events (not shown here). Stronger nutrient limitation all over the ocean counteracts the positive temperature effect on phytoplankton growth associated with MHWs." (l 418-420)

Apart from stronger nutrient limitation, no major differences were identified between the drivers of NPPX during simple NPPX events and during compound MHW-NPPX events.

**1.3   Specific Comments**

In addition to this I have a number of minor comments below that will hopefully improve manuscript clarity and accessibility. Subject to these comments being addressed, I am happy to recommend publication

L40 Is it fair to characterize 'the blob' as solely a MHW/NPPX event? What about cooccurring deoxygenation and acidification?

We replaced "It coincided with large negative anomalies in phytoplankton NPP [Whitney, 2015]." by "Along with anomalously low oxygen and high [H+] concentrations, the Blob coincided with large negative anomalies in phytoplankton NPP [Whitney, 2015, Gruber et al., 2021, Mogen et al., 2022]." (l. 43-44)

L74 A word seems missing after "phytoplankton". Growth, biomass, both?

We added the word "biology" after "phytoplankton".

L157 interval(s)

Thank you.

L161 Maybe clarify you mean "ocean ecology/biogeochemistry. The models presumably also have differing land ecosystem models/DGVMs.

We replaced "ecological module" by "ocean biogeochemical module".

L166 Are you sure this is correct? In the appendices it appears that MARBL uses higher half saturation constants. Maybe there's an error in the appendix legends or units. To aid comparison I would recommend using the same units for parameters that are present in each model where possible.
We now use the same units for parameters in each model. MARBL uses lower half saturation constants than TOPAZv2.

L167 Are nutrient levels similar between models? I ask because this is an interpretation that the authors come back to.

Indeed, background nutrient levels are different in the two models. For example, we compared surface phosphate concentration and found relatively smaller concentration in the low latitudes and higher concentration in the high latitudes in ESM2M-LE compared to CESM2-LE. Along with different half-saturation constants, different nutrient levels therefore explain divergent nutrient limitation in the two models.

We corrected any over-interpretation in the manuscript where we had by mistake assumed similar background nutrient levels. For example, we removed: "Assuming similar nutrient levels, this would imply lower nutrient limitation in CESM2." (section 2.2). We also revised the interpretation of divergent nutrient limitation in ESM2M-LE and CESM2-LE in section 3.3.1, where we replaced: "However, the models disagree on the strength of the nutrient limitation changes, potentially due to TOPAZv2 in ESM2M-LE using higher half-saturation constants than MARBL in CESM2-LE." by: "However, the models disagree on the strength of the nutrient limitation, potentially due a stronger reduction in nutrient levels in ESM2M-LE compared to CESM2-LE. Background nutrient

limitation is also higher in ESM2M-LE compared to CESM2-LE (Fig. B4 i-l) and therefore more sensitive to changes in nutrient levels (see the formulation of Nlim in sections A.1.2 and A.2.2)." (l. 422-426)

L182 Perhaps add something here on the rationale behind deseasonalizing. Do observations support this approach? It seems contrary to the thermal stress MHW estimates that are used for coral reefs.

In our study, we define extreme events relative to the seasonally-varying climatology. We therefore assume that marine ecosystems might be impacted from any extreme deviation from the seasonally-varying climatology. That may be the case for certain species, but certainly not for all species. Warm-water corals seem to be impacted by MHWs once a certain absolute temperature threshold is reached, e.g. 32°C for *Acropora* in northwest Australia [Schoepf et al., 2015].Although corals may motivate the use of an absolute temperature threshold for defining MHWs, we decided to use a relative threshold, following the common and widely-used definition of MHWs in recent literature (e.g. Hobday et al. [2016]). By doing so, MHWs can occur throughout the year and not only in summer, whereas NPPX events are not limited to the low productive seasons.

We added the following sentences to section 2.3:

"There are pros and cons to using a relative threshold compared to using an absolute threshold. Certain marine species might only be negatively impacted by MHWs and NPPX events once an absolute SST or NPP threshold is reached. Still, given our limited knowledge of marine ecosystems response to extremes, especially to NPPX events, we decided to align with the common definition of MHWs in recent literature, i.e., we define extreme events relative to the seasonal cycle [Hobday et al., 2016]. Thereby, we identify MHWs and NPPX events that would potentially impact all marine ecosystems vulnerable to extreme deviations from the seasonally-varying climatology." (l. 185 -190)

L186. Over what period are percentiles computed? Does this correspond to a 1-in-10-day event? This seems like a low threshold for defining an extreme that probably need to be discussed/justified.

We compute percentiles over the whole time series. Indeed, MHWs and NPPX events correspond to 1-in-10 day events. This low threshold allows for capturing enough compound MHW-NPPX events - whose expected frequency is 1-in-100 day. We added the following sentence to section 2.3:

"We use a relatively low threshold to define MHWs and NPPX events so as to capture enough compound MHW-NPPX events in the relatively short 1998-2018 time period over which NPP observations are available." (l. 194 -195)

L193. I'm not sure I would interpret these as hotspots of compound extremes, rather simply a lack of independence between the drivers of MHW and NPPX events. Warming/stratification/reduced nutrient supply?

We agree and replaced "Regions where their frequency exceeds 1% can be considered hotspots of unusually frequent compound MHW-NPPX events." by "Compound MHW-NPPX events can be considered unexpectedly frequent or unfrequent over all regions where their frequency is not equal to 1%, which indicates potential dependences between the drivers of MHWs and NPPX events." (l. 197-199)

L196 I find it a little strange to have model evaluation in the methods but I suppose there's a certain logic to not having this in the results.

In the model evaluation section, we evaluate the models performance at representing SST, NPP, MHWs and NPPX events. We placed it in the Methods so as to directly start the Results section with the analysis of compound MHW-NPPX events, which are the actual subject of this study. No changes are made to the manuscript.

Figure 1. Maybe mention that Std is normalized on the Taylor plot axes.

We changed the Taylor plot axes to "Normalized standard deviation".

Figure 1d. One wonders if an overestimation of durations is a resolution issue. E.g. a result of non-eddying models failing to capture short-lived extremes that may be associated with mesoscale processes.

That might indeed be the case, but gaps in satellite observations may also play a role. We added the following sentence to section 2.4: "Longer NPPX durations in ESM2M-LE compared to observations might arise from an overestimation of durations in the non-eddying ocean models, which might fail to capture short-lived extremes associated with mesoscale and submesoscale processes. However, it might also be explained by an underestimation of durations in the observations due to gaps in satellite observations." (l. 237-240)

L232 This may be an overstatement for NPP where the models seem a long way from the ref (even if there is very little confidence in the ref).

We replaced: "Overall, ESM2M-LE and CESM2-LE represent the mean state and variability of SST and NPP reasonably well and appear therefore suited to investigate the likelihood and drivers of compound MHW-NPPX events over the global ocean."

by: "Overall, ESM2M-LE and CESM2-LE represent the mean state and variability of SST reasonably well. Their representation of NPP diverges from observations, yet the reference for NPP observation in Figure 1 is computed as the mean of five observation-based NPP products which themselves disagree (Figure B1), although the spatial pattern of the mean state and variability of NPP is broadly similar across products. Considering that both ESM2M-LE and CESM2-LE capture this spatial pattern, they appear suited to investigate the likelihood and drivers of compound MHW-NPPX events over the global ocean." (l. 242-246)

L266 Is this advection/mixing term explicitly calculated or is this a residual? If the former can the authors say how well the decomposition works? Related to this, are the driver decompositions calculated online or using models outputs at a certain resolution?

The advection/mixing term is not explicitly calculated. As mentioned above we replaced "circulation" by "residual" in the revised manuscript. This residual includes the circulation contribution to driving biomass changes, as well as all errors resulting from low temporal resolution and vertical integration of variables. For example, the driver decomposition is computed using 5-day mean output. Equation 4 holds true at the time resolution of the model (i.e. 2 hour-time steps) only. A residual must be added to equation 4 when using 5-day mean outputs to account for non linearities. In the revised manuscript, we now consider the circulation term as a residual and removed all over-interpretations of the drivers of biomass changes during MHW-NPPX events.

L269 The method could be clarified here. Do you mean the maximum absolute anomaly? And if so, relative to what, the climatological mean?

This has been clarified. We write in section 2.5: "$n$ changes from its climatological mean value (at $t_1$) to its maximum absolute anomaly reached during a compound event (at $t_2$)." (l. 287-288)

Results- Lots of the interpretation and speculation that is currently in the results section is really "discussion". I would recommend having a combined results and discussion section to avoid substantial reformatting.

Indeed, there is some direct interpretation of our results in the Results section. We tried our best to remove all that had previously been discussed in the Results section, but decided to keep both sections. The Discussion goes beyond the interpretation of our study's results; it puts them into the perspective of the literature.

Figure 3. It's not entirely clear to me how to interpret the contours in this figure. Should there be values associated with the contours?

We decided to remove the contours as they do not provide much valuable information and might confuse readers.

L332. You're showing NPP not "abundance" (although they are presumably closely related in the climatological mean).

We replaced: "large phytoplankton NPP anomalies play an important role during MHW-NPPX events where the abundance of large phytoplankton is relatively high in the climatological mean."

by: "large phytoplankton NPP anomalies play an important role during MHW-NPPX events where the climatological mean state of large phytoplankton NPP generally dominates." (l. 358-359)

L345-347 I suggest removing the "contribution" phrasing here which is confusing.

We would rather keep the "contribution" phrasing, as the contributions of the growth rate anomaly $d\mu$ and of the biomass anomaly $dn$ are $nd\mu$ and $\mu dn$, respectively, as explained in the Methods section. The contributions $nd\mu$ and $\mu dn$ are not equivalent to $d\mu$ and $dn$. We now specify "(i.e., $nd\mu$)" in the manuscript (l. 372).

L356 Any such NPP increases are very difficult to see with the current color scale of Figure 3. Maybe you want to white-center your color scale to avoid overinterpretation of very low anomalies.

We white-centered Figures 3, 4 and 5 to facilitate the distinction between negative and positive anomalies. We also rescaled the color scale.

L364 Is the "per day" unit correct? I.e is this a rate of decline in nutrient limitation? I would expect a mean nutrient limitation anomaly to be unitless.

Yes, this is correct. Although, $T_f$, $L_{lim}$, and $N_{lim}$ are all unitless, their contributions to the growth rate anomaly $d\mu$ (day$^{-1}$) are in day$^{-1}$. As explained in section 2.5:

$d\mu$ can be decomposed into the contributions of a change in $T_f$, $L_{lim}$, and $N_{lim}$ during compound events.

$$d\mu_i = \mu_{max_i}(N_{lim_i}L_{lim_i}dT_{f_i} + N_{lim_i}T_{f_i}dL_{lim_i} + T_{f_i}L_{lim_i}dN_{lim_i}) \qquad (5)$$

$\mu_{max_i}$ is the maximum growth rate in day$^{-1}$. Therefore, the contribution of a change in the nutrient limitation to $d\mu$, $\mu_{max_i}T_{f_i}L_{lim_i}dN_{lim_i}$, is also in day$^{-1}$.

L373-374 I find it odd that the impact of nutrient limitation isn't exacerbated in the larger phytoplankton given their higher half saturation constants. Could the authors comment on this?

Higher half saturation constants in large phytoplankton compared to small phytoplankton result in a weaker decrease of $N_{lim}$ for a given reduction in nutrient concentration in a nutrient-limited regime. This is illustrated by Figure 3 below, which shows the phosphate limitation as a function of the phosphate concentration. Half saturation constants are taken from MARBL and correspond to 0.01 and 0.05 for small and large phytoplankton, respectively. Given the non linear response of the phosphate limitation to a change in the phosphate concentration, under limited nutrient conditions, a certain decrease in $[PO_4]$ actually results in a stronger decrease of $N_{PO_4}$ in small phytoplankton compared to large phytoplankton. That explains the exacerbated impact of nutrient limitation on smaller phytoplankton during compound events.

[Figure]

Figure 3: Phosphate limitation $N_{PO_4}$ against the $PO_4$ concentration for small and large phytoplankton in MARBL. The dashed vertical lines indicate the half saturation constants of small and large phytoplankton.

We added in section 3.3.1: "Divergent responses of the nutrient limitation to changes in nutrient levels during compound MHW-NPPX events for small and large phytoplankton can be explained by smaller half saturation constants in small phytoplankton, which, given the formulation of the nutrient limitation in MARBL (section A.2.2), would result in a stronger decrease of $N_{lim}$ given a certain decrease in nutrient levels. (l. 411-414)"

L366 As mentioned previously, why are MHW-NPPX events associated with enhanced light limitation in the high latitudes? Presumably these waters are more stratified or are sea-ice dynamics interfering with the expectation that mixing of phytoplankton below the euphotic zone declines during a MHW?

MHW-NPPX events are rare in the high latitudes. When they occur, they tend to be associated with light limitation on phytoplankton growth, which contributes to driving NPPX. On the other hand, univariate MHWs tend to be associated with increased radiative heating in ESM2M [Vogt et al., 2022], which has been identified as the main driver of MHWs in the high latitudes.

We added a paragraph to section 3.3.1 to explain how MHWs might co-occur with enhanced limitation in the high latitudes. See our response to the second main comment.

Figure 4. Perhaps the authors could add a comment in the legend on how well the growth rate decomposition works, assuming it is not perfect.

The decomposition of the growth rate anomaly (Fig. 4e-h) approximates the actual growth rate anomaly (Fig. 4a-d) during compound MHW-NPPX events with a certain residual (Fig. 4e-l), which is small in both models at global scale, yet remains substantial in certain regions (e.g., in the eastern equatorial Pacific) in CESM2-LE.

[Figure]

Figure 4: Growth rate anomaly relative to the climatological seasonal cycle during MHW-NPPX events for small (a) and large (b) phytoplankton in ESM2M2-LE and for small (c) and large (d) phytoplankton in CESM2-LE. Growth rate anomaly recomputed from its decomposition (e-h). Difference between the growth rate anomaly and its decomposition (i-l)

We added in the legend of Figure 4: "The decomposition of $d\mu$ into these three contributions comes with a global mean residual of 0.009 day$^{-1}$ and -0.002 day$^{-1}$ for small and large phytoplankton in ESM2M-LE, and of -0.007 day$^{-1}$ and 0.002 day$^{-1}$ for small and large phytoplankton in CESM2-LE."

L381 See earlier comment on this.

Please see earlier response.

Figure 5. The units in the figure and legend don't match.

Thank you, we corrected the units in the legend.

L421. Maybe clarify you mean the grazing of phytoplankton by zooplankton.

We replaced "grazing" by "grazing of phytoplankton by zooplankton".

Figure 6. I really like the idea behind this figure but it would be nice to see a sum of the contributions to see how well the decomposition works in each region. What does the "+res." term stand for (residual?). As previously stated, I am uncertain how to interpret these large circulation anomalies that offset NPP declines in all regions. Figure 6. I'm quite surprised by the very small Nlim term in CESM2, particularly in the low latitudes and the eastern equatorial Pacific. Can the authors comment on this.

We added a "residual" bar on Figure 6, which corresponds to the difference between dNPP and the contributions of the growth rate anomaly (via changes in Tf, Irrlim and Nlim) and the biomass anomaly to dNPP. We did not decompose the contribution of the biomass anomaly into the further contributions of NPP - Loss and circulation, as explained above.

The negative Nlim term on Fig. 6 indicates that nutrient limitation contributes to driving NPPX during MHW-NPPX events. However, this contribution is small in CESM2-LE, especially in low latitudes and the eastern equatorial Pacific compared to in ESM2M-LE. We added in section 3.3 that models "disagree on the strength of the nutrient limitation, especially in low latitudes and the eastern equatorial Pacific, potentially due to a stronger reduction in nutrient levels in ESM2M-LE compared to CESM2-LE. Background nutrient limitation is also higher in ESM2M-LE compared to CESM2-LE (Fig. B4 i-l) and therefore more sensitive to changes in nutrient levels (see the formulation of Nlim in sections A.1.2 and A.2.2)." (l. 422-426). Low background nutrient limitation in CESM2-LE (Fig. B4 k-l) signifies that very strong decreases in nutrient levels are needed to significantly raise the nutrient limitation, which might rarely happen. Thus, nutrient limitation does not seem to be the main driver of NPPX during MHW-NPPX events in CESM2-LE. Instead, we found reduced phytoplankton biomass to be the main driver. That could be because of a relative increase of phytoplankton grazing and mortality compared to phytoplankton production. Top down control might be a more potent driver of NPPX than nutrient limitation in CESM2-LE.

L443-444 Don't the models basically show no light limitation effect in the low latitudes? Perhaps best to refer to light limitation and not light levels. Presuming it's the vertical distribution of phytoplankton biomass and not incoming shortwave which is the main driver of light limitation during MHWs (unless authors have assessed this?).

We agree and replaced "The models agree on the sign of the light contribution in the low latitudes: enhanced light levels increase phytoplankton growth and thus moderate the negative NPP anomaly during MHW-NPPX events. Nevertheless, the models disagree in the high latitudes and in the equatorial Pacific." by "On average, in the low latitudes, changes in the light limitation hardly contribute to $dNPP$. In the high latitudes and in the equatorial Pacific, the models disagree on the sign of the light contribution." (l. 499-490)

L450 See previous comment.

Higher half-saturation constants imply a lower change in the nutrient limitation for a certain reduction in nutrient levels in a nutrient-limited regime. Therefore, we replaced: "However, they disagree on the strength of the nutrient limitation, potentially due to TOPAZv2 using higher half-saturation constants than MARBL." by "However, the models disagree on the strength of the nutrient limitation, potentially due a stronger reduction in nutrient levels in ESM2M-LE compared to CESM2-LE. Background nutrient limitation is also higher in ESM2M-LE compared to CESM2-LE (lower $N_{lim}$ in Fig. B4i-l) and therefore more sensitive to changes in nutrient levels (see the formulation of

$N_{lim}$ in sections A.1.2 and A.2.2)." (l. 422-426)

L457 Is there causality here? Do higher temperatures enhance loss or are they simply associated with enhanced loss?

Yes, there is causality. We write in section 3.3.2: "During MHWs, higher temperatures not only enhance NPP via their positive effect on the growth rate, they also directly enhance phytoplankton loss via their similarly positive effect on phytoplankton grazing and mortality (see Sections A.1.5 and A.2.5)." (l.459-461)

L478 I don't think this citing of the study "goals" is needed and is a bit distracting.

This citing is indeed redundant, yet we believe it might help readers that do not read the whole paper in one shot remember the study's objectives and follow the discussion's train of thoughts.

L496-497 Presumably nutrient and light limitation only influence production not loss. Can the authors discuss how loss is actually affected? Is there greater thermal sensitivity of grazers than phytoplankton? how do changes in physical dynamics influence loss during MHWs?

We replaced "Changes in the nutrient and light limitation may also be the reason why phytoplankton loss exceeds its production over the global ocean, resulting in the buildup of a negative biomass anomaly that contributes to driving the other half of the negative NPP anomaly during MHW-NPPX events."
by "Although higher temperature have the same enhancing effect on phytoplankton NPP and loss, nutrient and light limitation during MHW-NPPX events might decrease NPP sufficiently for it to be exceeded by phytoplankton loss over the global ocean. This relative increase of phytoplankton loss compared to NPP possibly explains the buildup of a negative biomass anomaly that contributes to driving the other half of the negative NPP anomaly during MHW-NPPX events." (l. 543-546)

As described above, we cannot say how ocean dynamics influence loss during MHW-NPPX events, as the difference between $\partial_t n_i$ and $NPP - Loss$ includes not only the circulation term but also all errors inherent to our assumption of equation **??** holding at a 5-day mean resolution and after averaging variables over the top 100-meter layer.

**2 Response to Anonymous Referee**

**2.1 General comments:**

This study aims at (1) evaluating the incidence of compound marine extreme events (high SST, low NPP) in two climate models and (2) identifying the drivers of these in these models, an impact task to be able to confidently project their changes in the future with these models. This study is interesting, very well written and address a very relevant scientific question within the journal's scope. I have a couple of major concerns that need to be addressed before being able to recommend its publication.

*We thank the reviewer for the encouraging feedback. We understand their concerns and have addressed them our best in this revised version of the manuscript.*

**2.2 Main comments:**

My major concerns are the following:

The role of ocean dynamics: if I understand it well, the authors derive the role of advective process from the difference between phytoplankton change from the beginning to the maximum of an NPPX event and the integral of (NPP – Loss) over the same period. While it appears sounded, I hardly believe the results displayed on Figure 5 and 6, where ocean dynamical processes almost systematically counteracts the impact of (NPP – Loss) for both models and in most regions. I suspect that there is a mistake in the conception or implementation of the method here. The only explanation the authors provide to explain this systematic behaviour is that the increased stratification reduces downward mixing and prevent the export of phytoplankton out of the top 100m. While this may be true in a limited number of cases, it is known that chlorophyll survives only in the euphotic layers and that a very large majority of the Chl concentration lies within the first 100m. Below these depths, Chl dies and this effect would be accounted in the Loss term rather in the ocean dynamics. Looking at Figure 5, it is obvious that the effect of (NPP-Loss) and dynamics clearly mirror each other, which I find very doubtful. This is particularly the case in the central/eastern equatorial Pacific, where dynamics act to systematically oppose the effect of (NPP-Loss). These events generally develop during El Nino events, during which equatorial divergence weakens and hence dynamics is supposed to systemically contribute to increase Chl biomass. It is clearly not the case for none of the models and Phytoplankton group as shown on Figure 6e,f. I also have a hard time to understand why the impact of circulation can be that different between small and large phytoplankton group, especially for ESM2M-LE model, where circulation changes are the same for both groups and Chl climatological distribution share the same patterns for these group (Figure B5i-l). These simple considerations led me to suspect some caveats in the computation in the dynamical contribution. I strongly recommend the authors to double check there method and, if they are convinced there is no mistake, discuss in more details the reasons that could lead to a systematic offset of (NPP-Loss) effect by the dynamics in the core of the result section but also in the conclusion, where this fact is never discussed.

In the Methods section, we reformulated the description of the drivers of changes in phytoplankton biomass (please see our response to Lester Kwiatkowski's first main comment 1.2). We conducted additional analyses to verify that all computations to obtain Figure 5 were correct.

For instance, we checked that biomass changes over time closely follow NPP-Loss. Figure 5 illustrates the biomass change of small and large phytoplankton at a grid cell in the northern Atlantic (47.5°N 30.5°W) for one LES (CESM2-LE). Indeed, $\partial_t n_i$ mirrors $NPP - Loss$, although $\partial_t n_i$ is usually lower than the $NPP - Loss$ contribution to driving $\partial_t n_i$. We had assumed the difference between $\partial_t n_i$ and $NPP - Loss$ to be $Circ$, the circulation contribution.

[Figure]

Figure 5: Contribution of biology to the biomass change of small and large phytoplankton at a grid cell in the northern Atlantic (47.5°N 30.5°W). The biomass change $\partial_t n_i$ (black line) closely follows the contribution of biology $NPP - Loss$ (blue line), explained by the difference between phytoplankton $NPP$ (green line) and $Loss$ (red line).

Our method to understanding the mean biomass anomaly during compound events was to integrate $\partial_t n_i$ and $NPP - Loss$ over periods over which the biomass anomaly builds up. However, when integrating, discrepancies between $\partial_t n_i$ and $NPP - Loss$ accumulate such that the $NPP - Loss$ contribution appears much larger than the observed biomass changes. In turn, the circulation contribution is large too and indeed systematically counteracts the impact of $NPP^\smile Loss$ for both models and in most regions.

However, the decomposition of $\partial_t n_i$ into a contribution from $NPP - Loss$ and $Circ$ (Equation 1) only holds at the time and vertical resolution of model computations, i.e, at 2-hour and 10-meter resolution. When using 5-day mean model output averaged over the top 100-meter layer, a residual term should be added to equation 1:

$$\partial_t n_i = NPP_i - Loss_i + Circ_i + Errors \tag{6}$$

We actually cannot differentiate between the circulation term and the errors. In the first version of this paper, we had assumed errors to be negligible and used equation 1 to explain biomass changes. However, they might be substantial. Given that we cannot assess how accurate equation 1 is, we revised over-interpretations of our results in the whole manuscript. The circulation term is now considered as a residual term, and the decomposition in equation 4 simply used as a tool to better understand the sign of biomass changes, rather than a way to quantify the exact contributions of biology and circulation to changes in biomass.

The role of light limitation: From Figure 4, the authors argue that light limitation is a major factor contributing to the growth rate anomalies in a lot of places, but they never provide a convincing explanation on how MHW could drive a change in light limitation. It is very surprising to see that the effect of light limitation are almost opposite to each other between the two models (Fig. 4ij compared to Figure 4kl). The authors hypothesize that this divergence may arise from different nutrient limitations, chlorophyll to carbon ratios and light harvest coefficients but it is very difficult to believe given the very similar formulation of L(lim) in the two models. I would have intuitively argued that changes in Irradiance would have played a major role. I would like the authors to further explore and discuss the mechanisms behind this very surprising and inconsistent role of light limitation in the revised manuscript.

We added the following paragraph to section 3.3.1:
"Even though the light limitation depends on a number of factors other than the light supply, such as temperature, nutrient levels, mixed layer depth or the carbon to chlorophyll ratio in phytoplankton, increased light limitation is here a direct result of reduced light supply by -13 $Wm^{-2}$ on average (Fig. 6a). High latitude MHWs are, however, mainly driven by enhanced shortwave radiation in summer (see Fig. 4 in Vogt et al. [2022]). Enhanced shortwave radiation seem incompatible with reduced light levels, hence the low compound MHW-NPPX event frequency in the high latitudes in ESM2M-LE. Therefore, for MHWs to co-occur with reduced light levels, they must be driven by other drivers than radiative heating, such as vertical mixing or advective processes. These drivers might be compatible with clouds or extended sea-ice cover, and thus with light limitation. In addition, high temperatures during MHWs also raise energy demand on phytoplankton and directly enhance the light limitation (see the role of $T_f$ in sections A.1.3 and A.2.3)." (l. 394-402)

We also conducted additional analyses to understand the divergent light limitations in the two models. Your intuition is correct. High latitude MHW-NPPX events are associated with enhanced light limitation due to reduced light levels in ESM2M-LE (Fig. 6a) and with reduced light limitation due to enhanced light levels in CESM2-LE (Fig. 6b).

[Figure]

Figure 6: Surface photosynthesis available radiation anomaly ( W m$^{-2}$) during MHW-NPPX events relative to the seasonal cycle in ESM2M-LE (a) and in CESM2-LE (b).

We added Fig. 6 in the Appendix of the manuscript and modified section 3.3.1 to clarify that changes in light levels contribute to changes in the light limitation. In particular, we added the following sentences:

"In ESM2M-LE, [...] increased light limitation is here a direct result of reduced light supply by -13 $Wm^{-2}$ on average (Fig. 6a)" (l.396),

"In CESM2-LE [...]. In the high latitudes, increased light levels by 7 $Wm^{-2}$ on average reduce light limitation (Fig. 6b), which ultimately enhances small and large phytoplankton growth" (l. 414-415), and

"the models disagree on their representation of the light limitation changes during MHW-NPPX events, especially in the high latitudes. This model divergence may arise from a number of factors involved in the calculation of $L_{lim}$, such as different light harvest coefficients in TOPAZv2 (Section A.1.3) and MARBL (Section A.2.3), but most importantly, divergent representation of the coupling between radiative fluxes, ocean temperature, and phytoplankton growth in the two models results in different light levels during MHW-NPPX events." (l. 426-430)

**2.3 Specific comments:**

Aside these two major points, find below a couple of minor comments that could improve the readability of the paper:

L161: These models not only differ in their ocean biogeochemical compartment but also in the physical ocean and atmosphere used, which could also explain some of the differences between the two models.

We added "Aside differences in their physical ocean and atmosphere modules, " (l. 159).

Figure 3: Different colorbars could be used for the two models to avoid saturation for CESM2-LE model (panels c,d,k,l)

To ease comparison between models, we decided to use a linear colorscale for both

ESM2M-LE and CESM2-LE. Avoiding saturation in the high northern latitudes in CESM2-LE would require very high limits of the colorbar ($> 600$mg C m$^{-2}$ day$^{-1}$), at the expense of pattern visibility elsewhere over the global ocean where absolute anomalies are usually lower than 200mg C m$^{-2}$ day$^{-1}$, especially on panels g,h. We thus decided to keep the same colorbars in ESM2M-LE and CESM2-LE, but to increase its limits to 300mg C m$^{-2}$ day$^{-1}$ to reduce saturation.

L385-389: Why not relating it to Irradiance changes?

In section 3.3.1, we now relate changes in the light limitation to changes in the light levels in each model. We also explain divergent light limitation in ESM2M-LE and CESM2-LE by different light levels during MHW-NPPX events.

L366: Provide more physical interpretation behind the increased light limitation.

Please see our response to comment 2.2.

Figure 5: Same as Figure 3. Use a different colorbar for the two models to avoid saturating colors for CESM2

We now use a different colorscale for the two models in Fig. 5. Limits of the colorbars have been extended to reduce saturation. In addition, we white-centered the colorscale in Figure 5 (as well as in Figure 3 and 4) to better distinguish between positive and negative anomalies

L411-418: This is the only place where the authors discuss the counteracting effect of dynamics over (NPP-Loss) and I don't find their explanation very convincing. If there is no mistake in the calculation, the authors definitely need to explain this systematic behaviour in a more convincing way...

Please see our response to comment 2.2. We removed the circulation term from Figure 5 and instead show the "residual" on a separate figure in the Appendix. This residual includes the circulation contribution to $dn$, as well as all errors inherent to our assumptions. We removed from the manuscript all over-interpretations regarding the contribution of ocean circulation to $dn$, as it cannot be differentiated from residual errors. Further studies are needed to better understand biomass changes, and especially the role of circulation to driving biomass changes.

**References**

N. Gruber, T. L. F. P. Boyd, and M. Vogt. Ocean biogeochemical extremes, compound events. *Nature*, 2021.

A. J. Hobday, L. V. Alexander, S. E. Perkins, D. A. Smale, S. C. Straub, E. C. Oliver, J. A. Benthuysen, M. T. Burrows, M. G. Donat, M. Feng, N. J. Holbrook, P. J. Moore, H. A. Scannell, A. Sen Gupta, and T. Wernberg. A hierarchical approach to defining marine heatwaves. *Progress in Oceanography*, 141:227 – 238, 2016. ISSN 0079-6611. doi: https://doi.org/10.1016/j.pocean.2015.12.014. URL `http://www.sciencedirect.com/science/article/pii/S0079661116000057`.

S. C. Mogen, N. S. Lovenduski, A. R. Dallmann, L. Gregor, A. J. Sutton, S. J. Bograd, N. C. Quiros, E. Di Lorenzo, E. L. Hazen, M. G. Jacox, M. P. Buil, and S. Yeager. Ocean biogeochemical signatures of the north pacific blob. *Geophysical Research Letters*, 49 (9):e2021GL096938, 2022. doi: https://doi.org/10.1029/2021GL096938.

V. Schoepf, M. Stat, J. Falter, and M. McCulloch. Limits to the thermal tolerance of corals adapted to a highly fluctuating, naturally extreme temperature environment. *Scientific Reports*, 5:17639, 12 2015. doi: 10.1038/srep17639.

L. Vogt, F. A. Burger, S. M. Griffies, and T. L. Frölicher. Local drivers of marine heatwaves: A global analysis with an earth system model. *Frontiers in Climate*, 4, 2022. ISSN 2624-9553. doi: 10.3389/fclim.2022.847995.

F. A. Whitney. Anomalous winter winds decrease 2014 transition zone productivity in the ne pacific. *Geophysical Research Letters*, 42(2):428–431, 2015. doi: 10.1002/2014GL062634.

---

## Referee Report (RR1)

Second round of review of "Hotspots and drivers of compound marine heatwave and low net primary production extremes" by N. Le Grix et al.

The authors did a nice work to account for most of my major concerns in intial evaluation of the paper. There are however two changes that I could like the authors to consider (and some very minor comments) before the paper can be definitely suitable for publication.

My first major comments relates again to their decomposition procedure. The authors indeed writes L295: $dn_i \sim \Delta n_i$. If I understand it well:

- $dn_i$ refers to mean biomass anomaly between $t_0$ (climatological value) and $t_{max}$ (maximum absolute anomaly relative to climatology)
- $\Delta n_i$ refers to the integrated biomass change between $t_0$ and $t_{max}$

Given the formula, $\Delta n_i$ is thus simply the difference in biomass between $t_0$ and $t_{max}$ (i.e. $ni(t_{max}) - ni(t_0)$), while dni is the biomass anomaly averaged between $t_0$ and $t_{max}$ (i.e. $\Sigma dn_i$). By contruction, $\Delta n_i$ should therefore be systematically larger than $dn_i$ (about twice larger), which indeed appears to be the case on Figure 5.

By definition,

$dni = (dni(t_0) + dni(t_1) + dni(t_2) + ... + dni(t_{max}))/N$, N being the number of 5-days output timesteps between $t_0$ and $t_{max}$

Thus:

$dn_i = (\partial_t n_i(t_0)* \Delta t+(\partial_t n_i(t_0)+\partial_t n_i(t_1))*\Delta t+(\partial_t n_i(t_0)+\partial_t n_i(t_1)+\partial_t n_i(t_2))*\Delta t+ ...+(\Sigma\partial_t n_i(t_0 ->t_{max})* \Delta t))/N$

$dn_i =( (N*\partial_t n_i(t_0)+(N-1)* \partial_t n_i(t_1)+(N-2)* \partial_t n_i(t_2)+...+ 1*\partial_t n_i(t_{max}))* \Delta t)/N$

As you can see, this calculation is clearly different from $\Sigma\partial_t n_i(t_0 ->t_{max})* \Delta t$, which corresponds to your definition of $dn_i$

I would either recommand calculating:

$\Delta n_i = (\partial_t n_i(t_0)* \Delta t+(\partial_t n_i(t_0)+\partial_t n_i(t_1))*\Delta t+(\partial_t n_i(t_0)+\partial_t n_i(t_1)+\partial_t n_i(t_2))*\Delta t+ ...+(\Sigma\partial_t n_i(t_0 ->t_{max})* \Delta t))/N$ (and the corresponding contributions) and compare it to the $dn_i$ as defined in the manuscript

Or redefine $dn_i$ as $n_i(t_{max}) - n_i(t_0)$ and compare it to $\Delta n_i$ as defined in the manuscript.

There will otherwise by a mathematical inconsistency and I suspect that proceeding either ways will end up in a closer match between $\Delta n_i$ and $dn_i$ in the paper.

My second comment relates to the fact that the authors do not discuss anywhere the fact that their biological contribution systematically exceeds the integrated biomass changes, i.e. that the residual (that the authors previously attributed to ocean dynamics)

systemically opposes the biological contribution. I would recommend the authors adding a small paragraph in the discussion section where they could provide hypothesis to explain this behaviour (that I still don't really understand).

Minor comments:

L553: "(see section 2.5," : a closing parathesis is missing.

L553: "resulting in in": remove one "in"

---

## Author Response (AR2)

**Hotspots and drivers of compound marine heatwave and low net primary production extremes**
**Response to the second round of reviews by referee 2**

Natacha Le Grix, Jakob Zscheischler, Keith B. Rodgers,
Ryohei Yamaguchi, and Thomas L. Frölicher

November 10, 2022
* * *
The authors did a nice work to account for most of my major concerns in initial evaluation of the paper. There are however two changes that I could like the authors to consider (and some very minor comments) before the paper can be definitely suitable for publication.

We thank the reviewer for their feedback on our previous response and their in-depth reviewing of our Methods.

My first major comments relates again to their decomposition procedure. The authors indeed write (l. 295): $dn_i \sim \Delta n_i$. If I understand it well:

- $dn_i$ refers to mean biomass anomaly between $t_0$ (climatological value) and $t_{max}$ (maximum absolute anomaly relative to climatology).

There might be a misunderstanding here. We define $dn$ as the mean biomass anomaly over all compound event days. Using the reviewer's notation, $t_0$ would be the first compound event day in the time series, when both SST and NPP exceed their extreme event thresholds, and $t_{max}$ the last compound event day in the time series.

- $\Delta n_i$ refers to the integrated biomass change between $t_0$ and $t_{max}$.

That is correct. In our manuscript, we had defined $\Delta n_i$ as the integrated biomass change between $t_1$ (climatological value) and $t_2$ (maximum absolute anomaly relative to climatology). We like the reviewer's use of $t_0$ and $t_{max}$ and replace $t_1$ by $t_0$ and $t_2$ by $t_{max}$ in the revised manuscript:

$\Delta n_i$ corresponds to the integrated biomass change over the period over which the biomass changes from its climatological mean value (at $t_0$, $n_i(t_0) = 0$) to its maximum absolute anomaly reached during a compound event (at $t_{max}$, $n_i(t_{max}) = n_{imax}$).

$$\Delta n_i = \int_{t_0}^{t_{max}} \partial_t n_i dt = n_i(t_{max}) - n_i(t_0) = n_{imax} \tag{1}$$

Given the formula, $\Delta n_i$ is thus simply the difference in biomass between $t_0$ and $t_{max}$ (i.e. $\mathrm{n}_i(t_{max}) - n_i(t_0)$), [...]

That is correct (see equation 1).

[...] while $dn_i$ is the biomass anomaly averaged between t0 and tmax (i.e. $\sum dn_i$).

As stated above, this is not correct. We rather define $dn_i$ as the biomass anomaly averaged over all compound event days.

By construction, $\Delta n_i$ should therefore be systematically larger than $dn_i$ (about twice larger), which indeed appears to be the case on Figure 5.

$\Delta n_i$ is the mean biomass change between $t_0$ and $t_{max}$ for all compound events, which is equivalent to the mean largest biomass anomaly $n_i(t_{max})$ during compound events. $dn_i$ is the mean biomass anomaly over all compound event days. We agree that $\Delta n_i$ is, by construction, expected to be systematically larger than $dn_i$, which indeed appears to be the case on Figure 5.

By definition,

$dn_i = (dn_i(t_0) + dn_i(t_1) + dn_i(t_2) + ... + dn_i(t_{max}))/N$, N being the number of 5-days output timesteps between $t_0$ and $t_{max}$

As explained above, we define $dn$ as the mean biomass anomaly over all compound event days (and not the mean biomass anomaly between $t_0$ and $t_{max}$).
$dn = (dn_i(t_{first}) + dn_i(t_{second}) + ... + dn_i(t_{last}))/N^*$, $N^*$ being the number of compound event days in the time series.

Thus:

$\mathrm{dn}_i = (\partial_t n_i(t_0)\Delta t + (\partial_t n_i(t_0) + \partial_t n_i(t_1))\Delta t + (\partial_t n_i(t_0) + \partial_t n_i(t_1) + \partial_t n_i(t_2))\Delta t + ... + \sum \partial_t n_i(t_0-> t_{max})\Delta t)/N$

$dn_i = ((N * \partial_t n_i(t_0) + (N-1)\partial_t n_i(t_1)) + (N-2)\partial_t n_i(t_2)) + ... + 1 * \partial_t n_i(t_{max}))\Delta t)/N$

As you can see, this calculation is clearly different from $\sum \partial_t n_i(t_0-> t_{max})\Delta t$, which corresponds to your definition of $dn_i$.

We thank the reviewer for writing these detailed calculations. However, we define $dn$ as the mean biomass anomaly over all compound event days. Please note that the first compound event day in the time series may follow a day of anomalous biomass, and that compound event days are not necessarily in succession. The above calculations cannot compute $dn$. They would have worked well, however, to compute the mean biomass

anomaly between $t_0$ and $t_{max}$.

I would either recommend calculating:

$\Delta n_i = (\partial_t n_i(t_0)\Delta t + (\partial_t n_i(t_0) + \partial_t n_i(t_1))\Delta t + (\partial_t n_i(t_0) + \partial_t n_i(t_1) + \partial_t n_i(t_2))\Delta t + ... + \sum \partial_t n_i(t_0 - > t_{max})\Delta t)/N$ (and the corresponding contributions) and compare it to the dni as defined in the manuscript.

We tried following the suggestion to redefine $\Delta n_i$ as the mean biomass anomaly between $t_0$ and $t_{max}$. We recomputed $\Delta n$ and its corresponding contributions as the cumulative sum of $\partial_t n_i \Delta t$, of $NPP\Delta t$, and of $Loss\Delta t$, divided by the number of time steps between $t_0$ and $t_{max}$. However, the recomputed $\Delta n$ was not as good an approximation of $dn$ as the $\Delta n$ we had defined in the manuscript (see Fig. 1a-h compared to Fig. 5a-h in the manuscript). The recomputed $\Delta n$ underestimates $dn$ (Fig. 1e-h compared to Fig. 1a-d). Therefore, we stick to our previous definition: $\Delta n$ remains the integrated biomass change between $t_0$ and $t_{max}$.

[Figure]

Figure 1: Biomass anomaly $dn$ (mg C m$^{-2}$) of small and large phytoplankton during compound MHW-NPPX events in ESM2M-LE (a,b) and in CESM2-LE (c,d). Mean biomass anomaly $\Delta n$ (mg C m$^{-2}$) over the period over which biomass changes from its climatological mean value up to its maximum anomaly reached during a compound MHW-NPPX event (e-h). Contribution of biological processes ($NPP - Loss$, i-l) to $\Delta n$.

Or redefine $dn_i$ as $n_i(t_{max}) - n_i(t_0)$ and compare it to $\Delta n_i$ as defined in the manuscript.

We decided not to follow the suggestion to recompute $dn_i$ as $n_i(t_2) - n_i(t_1)$. $dn$ remains computed as the mean biomass anomaly during compound event days, so as to stay consistent with the rest of the manuscript, where we decompose $dNPP$ into the contributions of the growth rate anomaly $d\mu$ and of the biomass anomaly $dn$. Note that $\Delta n_i$, the integrated biomass change between $t_1$ and $t_2$, is itself equal to $n_i(t_2) - n_i(t_1)$ (equation 1).

There will otherwise by a mathematical inconsistency and I suspect that proceeding either ways will end up in a closer match between $\Delta n_i$ and $dn_i$ in the paper.

We explain above that our definition of $\Delta n_i$ yields a better estimate of $dn_i$, and that we do not change the definition of $dn_i$.

We agree that $dn$ and $\Delta n$ are, by definition, different and removed "$dn_i \sim \Delta n_i$" in equations 7 and 8. Still, one can apprehend the sign of $dn$ by the build up of a compound event's maximum absolute biomass anomaly, i.e., by $\Delta n$. On Figure 5 of the manuscript, panels a-b and e-h are similar, which supports our method of using $\Delta n$ to gain a first understanding of $dn$.

We clarified the difference between $dn$ and $\Delta n$ in section 2.5 where we write: "Over time, biomass changes build up a biomass anomaly $dn$ that might be sufficient to drive or contribute to driving extremely low $dNPP$. In this study, we intend to explain the contribution of $dn$ to $dNPP$ during compound MHW-NPPX events using equation 6. A positive or negative biomass anomaly during a compound event may be explained by an overall increase or decrease in biomass over time, until the largest biomass anomaly reached during the compound event. Therefore, we integrate $\partial_t n$, $NPP$ and $Loss$ over all periods over which $dn$ builds up, i.e., over which $n$ changes from its climatological mean value (at $t_0$, $n_i(t_0) = 0$) to its maximum absolute anomaly relative to the climatology reached during a compound event (at $t_{max}$, $n_i(t_{max}) = n_{imax}$). $\Delta n$ refers to the integrated biomass change between $t_0$ and $t_{max}$, which corresponds to the largest biomass anomaly reached during a compound event. Note that $\Delta n$ is not exactly equivalent to $dn$. $\Delta n$ is a tool to understand the build-up of the largest biomass anomaly reached during a compound event, whereas $dn$ is the mean biomass anomaly over all compound event days." (l. 287-294)

Lastly, we replaced "this method would enable us to quantify the contribution of biological processes to $dn$." by "this method would enable us to apprehend the contribution of biological processes to $dn$." (l. 299)

My second comment relates to the fact that the authors do not discuss anywhere the fact that their biological contribution systematically exceeds the integrated biomass changes, i.e. that the residual systematically opposes the biological contribution. I would recommend the authors adding a small paragraph in the discussion section where they could provide hypotheses to explain this behaviour (that I still don't understand).

We added to Section 6: "Note that integrated $NPP$ - $Loss$ generally exceeds the integrated biomass changes (Fig. 5e-l), with some exceptions, e.g., in the high latitudes for small phytoplankton in ESM2M-LE. $\Delta n$, $NPP$ and $Loss$ terms include an error term when computed from 5-day mean, 10-meter vertically integrated biomass. Further studies at higher temporal and vertical resolution are needed to remove errors in all terms in equation 8, so as to quantify the exact $NPP$ - $Loss$ contribution to $\Delta n$." (l. 462-465). As we do not not know these error terms, we do not want to speculate reasons for the $NPP - Loss$ contribution usually exceeding the integrated biomass changes.

Minor Comments:

L553: "(see section 2.5,": a closing parathensis is missing.

We added a closing parenthesis.

L553: "resulting in in": remove one "in"

We removed an extra "in".